# Investigation of Arctic middle-atmospheric dynamics using 3 years of $H_2O$ and $O_3$ measurements from microwave radiometers at Ny-Ålesund

Franziska Schranz[1], Brigitte Tschanz[1], Rolf Rüfenacht[1,a], Klemens Hocke[1], Mathias Palm[2], and Niklaus Kämpfer[1]

[1]Institute of Applied Physics, University of Bern, Bern, Switzerland
[2]Institute of Environmental Physics, University of Bremen, Bremen, Germany
[a]now at: Federal Office for Meteorology and Climate MeteoSwiss, Payerne, Switzerland

**Correspondence:** Franziska Schranz (franziska.schranz@iap.unibe.ch)

**Abstract.**

We used 3 years of water vapour and ozone measurements to study the dynamics in the Arctic middle atmosphere. We investigated the descent of water vapour within the polar vortex, major and minor sudden stratospheric warmings and periodicities at Ny-Ålesund. The measurements were performed with the two ground-based microwave radiometers MIAWARA-C and GROMOS-C which are co-located at the AWIPEV research base at Ny-Ålesund, Svalbard (79° N, 12° E) since September 2015. Both instruments belong to the Network for the Detection of Atmospheric Composition Change (NDACC). The almost continuous datasets of water vapour and ozone are characterised by a high time resolution in the order of hours. A thorough intercomparison of these datasets with models and measurements from satellite, ground-based and in-situ instruments was performed. In the upper stratosphere and lower mesosphere the MIAWARA-C water vapour profiles agree within 5 % with SD-WACCM simulations and ACE-FTS measurements on average whereas AuraMLS measurements show an average offset of 10–15 % depending on altitude but constant in time. Stratospheric GROMOS-C ozone profiles are on average within 6 % of the model SD-WACCM, the satellite instruments AuraMLS and ACE-FTS and the ground-based microwave radiometer OZORAM which is also located at Ny-Ålesund.

During these first three years of the measurement campaign typical phenomena of the Arctic middle atmosphere took place and we analysed their signatures in the water vapour and ozone measurements. Two major sudden stratospheric warmings (SSWs) took place in March 2016 and February 2018 and three minor warmings were observed in early 2017. Ozone rich air was brought to the pole and during the major warmings ozone enhancements of up to 4 ppm were observed. The reversals of the zonal wind accompanying a major SSW were captured in the GROMOS-C wind profiles which are retrieved from the ozone spectra. After the SSW in February 2018 the polar vortex re-established and the water vapour descent rate in the mesosphere was 355 m/day. Inside of the polar vortex in autumn we found the descent rate of mesospheric water vapour from MIAWARA-C to be 435 m/day on average. We find that the water vapour descent rate from SD-WACCM and the vertical velocity $\overline{w}^*$ of the residual mean meridional circulation from SD-WACCM are substantially higher than the descent rates of MIAWARA-C. $\overline{w}^*$ and the zonal mean water vapour descent rate from SD-WACCM agree within 10 % after the SSW whereas in autumn $\overline{w}^*$ is up

to 40 % higher. We further present an overview of the periodicities in the water vapour and ozone measurements and analysed seasonal and interannual differences.

## 1 Introduction

In the Arctic middle atmosphere the solar irradiation conditions change dramatically throughout the year. These seasonal
insolation changes drive the typical summer and winter pattern in the atmospheric dynamics and the photochemistry of trace species in the polar stratosphere and mesosphere. In winter, during the polar night, middle atmospheric air is descending and a cyclonic wind system, the polar vortex, dominates the middle atmospheric dynamics and builds a mixing barrier between Arctic and midlatitude air. In summer, when the sun never sets, airmasses are rising and an anticyclone forms around the pole, which is however much weaker than the winter polar vortex.

Ny-Ålesund is located at 79°N and is therefore the ideal location to perform observations in the polar middle atmosphere. In winter Ny-Ålesund is mostly located inside of the polar vortex but the vortex centre shifts away from the pole frequently during midwinter and therefore observations in and outside of the polar vortex are possible. Many phenomena related to the dynamics of the polar vortex can be observed from this high latitude location. The most dramatic events which occur in the polar winter atmosphere are sudden stratospheric warmings (SSW). These events couple all atmospheric layers (Baldwin and Dunkerton,
2001; Funke et al., 2010) and lead to temperature increases of more than 25 K within a few days and zonal wind reversals in the stratosphere, whereas the temperature decreases in the mesosphere. The polar vortex is thereby shifted or even splits in two or more sub-vortices. Global-scale planetary waves can interact with the mean flow in the middle atmosphere and influence the large scale circulation. The wave–mean-flow interaction is believed to be the main cause of SSWs (Matsuno, 1970; Liu and Roble, 2002). Within the winter polar vortex mesospheric and stratospheric air descends and transports chemical constituents
to lower altitudes. This includes long lived constituents like NOx and HOx which lead to catalytic ozone destruction in the mesosphere and stratosphere (eg.  Randall et al., 2009). Through this mechanism energetic particle precipitation, which produces NOx and HOx in the mesosphere and lower thermosphere, has an influence on polar ozone (Andersson et al., 2018). In this article we focus on the analysis of dynamical events in the polar middle atmosphere as seen from Ny-Ålesund using water vapour and ozone data from ground-based microwave radiometers.

Water vapour is a valuable tracer for transport in the Arctic middle atmosphere. It has a lifetime in the order of weeks in the upper mesosphere and in the order of months in the lower mesosphere (Brasseur and Solomon, 2005) which is long compared to the time scales of dynamical processes. Additionally it has vertical and horizontal gradients in its volume mixing ratio (VMR). In general in the stratosphere water vapour is increasing with altitude because of water vapour production through photodissociation of methane leading to a positive vertical gradient. In the mesosphere the vertical gradient is negative because
water vapour is photodissociated through the absorption of Lyman-$\alpha$ radiation (Brasseur and Solomon, 2005). The horizontal gradients occur in winter. Airmasses descend within the polar vortex and in the absence of solar irradiation the water vapour maximum descends with it. This leads to a negative gradient across the vortex edge in the mesosphere and a positive gradient in the lower and middle stratosphere (Lossow et al., 2009; Maturilli et al., 2006). Water vapour has previously been used as a

tracer to study dynamical events like periodicities, transport during SSWs and descent rates within the polar vortex (Tschanz and Kämpfer, 2015; Bailey et al., 2014; Straub et al., 2012; Scheiben et al., 2012; Lee et al., 2011).

Stratospheric ozone can be used as a tracer of horizontal transport processes in winter where odd oxygen is under dynamical control and ozone has a strong gradient across the polar vortex edge. It can therefore be used to distinguish air parcels from in and outside of the polar vortex. Together with water vapour it was used as a tracer for vortex filamentation in the lower stratosphere (Müller et al., 2003) and for streamers of enhanced ozone in the middle stratosphere along the edge of the polar vortex over the Atlantic sector (Hocke et al., 2017).

Ground-based microwave radiometry is the ideal technique to monitor water vapour and ozone in the middle atmosphere. It allows a continuous observation under all weather conditions except during rain and has a high time resolution in the order of hours. The two ground-based microwave radiometers GROMOS-C for ozone and MIAWARA-C for water vapour are specially designed for campaigns and are therefore compact, easy to maintain and remote controlled. Since September 2015 the two instruments are located at the AWIPEV research base at Ny-Ålesund, Svalbard. In the Arctic MIAWARA-C is the only ground-based instrument continuously measuring middle atmospheric water vapour profiles except for the radiometer Vespa22 (Mevi et al., 2018) located at Thule, Greenland. Ozone profiles in the Arctic middle atmosphere are measured with GROMOS-C and with OZORAM (Palm et al., 2010a) which has been located at Ny-Ålesund since 1993. During summer also the FTIR (Palm et al., 2010b) provides middle atmospheric ozone profiles. The additional benefit of GROMOS-C at Ny-Ålesund is that it provides ozone measurements down to about 20 km and zonal and meridional wind profiles and it can switch the frequency to measure carbon monoxide.

In this article we present 3 years of almost continuous water vapour and ozone volume mixing ratio (VMR) measurements in the middle atmosphere above Ny-Ålesund (79°N) in the Arctic. The measurements were performed with the two ground based microwave radiometers MIAWARA-C and GROMOS-C. The high time resolution and the continuous measurements allow us to investigate the Arctic dynamics from diurnal to interannual time scales and to investigate different aspects of the Arctic dynamics. We present estimates of the water vapour descent rate inside of the polar vortex and discuss its meaning using the residual mean meridional circulation calculated from SD-WACCM, investigate water vapour and ozone changes and airmass exchange during sudden stratospheric warmings and analyse the seasonal and interannual variations of normal mode Rossby waves over these 3 years and compare it to models. Additionally we perform a thorough intercomparison with a model, a reanalysis, satellite measurements and co-located ground-based and in-situ measurements from Ny-Ålesund.

The remainder of this article is organised as follows: An introduction to the campaign at Ny-Ålesund is given in Sect. 2. In Sect. 3 the ground based microwave radiometers MIAWARA-C and GROMOS-C are described in detail as well as the instruments and models used for intercomparison and the methods are explained in Sect. 4. The water vapour and ozone datasets of the microwave radiometers MIAWARA-C and GROMOS-C are described in Sect. 5 and a comprehensive intercomparison with other instruments and models is presented in Sect. 6. In the following sections we report on typical phenomena of the Arctic middle atmosphere as observed in the water vapour, ozone and wind time series from Ny-Ålesund: the descent of air in the polar vortex during its formation (Sect. 7), major and minor sudden stratospheric warmings (Sect. 8) and periodicities (Sect. 9). Summary and conclusion are given in Sect. 10.

## 2 Ny-Ålesund Campaign

Within the frame of the Ny-Ålesund campaign MIAWARA-C and GROMOS-C, two ground-based microwave radiometers of the University of Bern, were moved to the AWIPEV research base at Ny-Ålesund/Svalbard (79° N, 12° E) in September 2015. Currently the instruments have been measuring water vapour and ozone VMR profiles at Ny-Ålesund for 3 years and the campaign is ongoing. This campaign is a collaboration with the University of Bremen which operates the ozone radiometer OZORAM at Ny-Ålesund since 1994. All instruments belong to the Network for the Detection of Atmospheric Composition Change (NDACC, De Mazière et al., 2018).

The aim of the campaign is to study the variability and possible anti-correlation between water vapour and ozone and to investigate dynamical and chemical phenomena in the polar middle atmosphere. Diurnal ozone variations and their seasonal differences have already been investigated (Schranz et al., 2018). In this article we present the water vapour and the ozone datasets of MIAWARA-C and GROMOS-C and we concentrate on analysing the traces of dynamical events in both water vapour and ozone. A possibility for the future would be to investigate spatial ozone variations from the measurements in the four cardinal directions or the effects of energetic particle precipitation on the water vapour and ozone concentration.

## 3 Instruments and Models

### 3.1 MIAWARA-C

MIAWARA-C, the MIddle Atmospheric WAter vapour RAdiometer for Campaigns, is a ground-based microwave radiometer built at the University of Bern and specially designed for campaigns. It is therefore a very compact instrument which only needs a power- and an internet-connection and which is operated remotely. The instrument front end is an uncooled heterodyne receiver with a system temperature of 150 K. In the back end the signal is spectrally analysed with an FFT spectrometer with 400 MHz bandwidth and 30.5 kHz spectral resolution. The instrument measures the pressure broadened emission line of water vapour at 22 GHz. The retrieval of the water vapour profiles from the spectra is performed with QPACK (Eriksson et al., 2005) and ARTS2 (Eriksson et al., 2011), using an optimal estimation method (Rodgers, 1976). An a priori water vapour profile is required for the optimal estimation method and is taken from an MLS climatology of the years 2004–2008. The retrieved water vapour profiles have an altitude range of 37–75 km with a vertical resolution of 12–19 km. For MIAWARA-C retrievals with a constant time resolutions (6h, 12h and 24h) and with a constant noise level of 0.014 K are performed. For the constant noise level retrieval the integration time depends on the tropospheric opacity and ranges from 30 minutes to more than 1 day. For 80 % of the retrievals the integration time is below 2 hours. Detailed descriptions of the instrument can be found in Straub et al. (2010) and Tschanz et al. (2013).

MIAWARA-C has been located at Bern (47° N, 7°E) and Sodankylä (67° N, 27° E) in the years 2010–2013 where a comprehensive intercomparison was performed and the mean bias to satellites and other ground-based microwave radiometers was calculated (Tschanz et al., 2013). With respect to MLS version 3.3, there is almost no bias at the lowest altitude level (4

hPa) but with higher altitudes the bias increases and at 0.02 hPa MIAWARA-C measures up to 13 % less water vapour than MLS.

## 3.2 GROMOS-C

GROMOS-C, the GRound-based Ozone MOnitoring System for Campaigns, is a microwave radiometer built at the University of Bern. Similarly to MIAWARA-C it is a campaign instrument and the design is very compact. It needs a power- and an internet-connection and is then controlled remotely. The instrument has an uncooled heterodyne receiver system with a system temperature of 1080 K. The spectral analysis of the signal is performed with a FFT spectrometer of 1 GHz bandwidth and a spectral resolution of 30.5 kHz. In its basic mode GROMOS-C measures the pressure broadened emission line of ozone at 110.8 GHz. It can however switch to measure the CO-line at 115.3 GHz. Additionally it is possible to retrieve wind profiles from the ozone spectra according to the measurement principle of Rüfenacht et al. (2012) and Rüfenacht and Kämpfer (2017). Therefore GROMOS-C observes subsequently in the four cardinal directions (N-E-S-W) at 22° elevation with a sampling time of 4 seconds. At the Arctic location of Ny-Ålesund this allows observations in and outside of the polar vortex if the vortex edge is close to Ny-Ålesund (Schranz et al., 2018). The retrieval of ozone and wind is performed with Qpack (Eriksson et al., 2005) and ARTS2 (Eriksson et al., 2011), using an optimal estimation method (Rodgers, 1976). For ozone the a priori is taken from an MLS climatology of the years 2004–2013 whereas a zero a priori with relatively large variance is used for wind. The ozone profiles of GROMOS-C have an altitude range of 20–70 km with a vertical resolution of 10–12 km in the stratosphere and up to 20 km in the mesosphere. The time resolution is 2 hours. The wind profiles range from about 40 up to 60–70 km with a vertical resolution of 10–20 km. Detailed information about the instrument can be found in Fernandez et al. (2015).

A validation campaign took place at La Reunion in 2014 and revealed that GROMOS-C and MLS ozone measurements agree within 5 % up to 0.2 hPa. A comparison with the ground-based wind radiometer WIRA (Rüfenacht et al., 2014) at La Reunion shows that GROMOS-C captures the principal wind features such as the stratospheric wind reversal in July 2014 (Fernandez et al., 2016).

## 3.3 OZORAM

OZORAM, the OZone Radiometer for Atmospheric Measurements, is a ground-based microwave radiometer built at the University of Bremen. The instrument is located at Ny-Ålesund since 1994 and is in its current observation mode since 2008. It has a cooled heterodyne receiver system with a system noise temperature of 1400 K. The signal is then spectrally analysed with an FFT spectrometer with 800 MHz bandwidth and a spectral resolution of 60 kHz. OZORAM measures the pressure broadened ozone emission line at 142 GHz. For the retrieval of the ozone profiles from the measured spectra ARTS 1.1 (Buehler et al., 2005) is used together with QPACK (Eriksson et al., 2005). The profiles have an altitude range of 30–70 km with a vertical resolution of 10–20 km and a time resolution of 1 hour.

A detailed description of the instrument can be found in Palm et al. (2010a) where also an intercomparison with the satellite instruments MLS and SABER was performed. OZORAM is within 10 % of MLS in the stratosphere and within 30 % of SABER in the mesosphere.

## 3.4 EOS-MLS

The Microwave Limb Sounder (MLS) is an instrument on board NASA's EOS-Aura satellite which was launched in 2004 (Waters et al., 2006). The satellite is in a sun synchronous orbit at 705 km altitude and with 98° inclination. It passes at Ny-Ålesund two times a day at around 04:00 and 10:00 UTC. We used water vapour (Lambert et al., 2015), ozone (Schwartz et al., 2015a) and temperature (Schwartz et al., 2015b) measurements. Water vapour is retrieved from the 183 GHz line and ozone from the 240 GHz band. Temperature is derived from radiances measured by the 118 and 240 GHz radiometers. It provides ozone profiles from 12–80 km altitude, water vapour from 5–80 km altitude and temperature from 10–90 km. The vertical resolution is 2.7–6 km. Profiles for comparison are selected if their location is within $\pm 1.2°$ latitude and $\pm 6°$ longitude from Ny-Ålesund. For this study we use the retrieval version 4.2.

In the SPARC water vapour assessment (Nedoluha et al., 2017) the MLS water vapour data set was intercompared to ground-based microwave radiometers from all over the world. The study showed that MLS water vapour profiles are typically 0-10 % higher than the profiles from the microwave radiometers in the range of 3–0.03 hPa. MLS ozone profiles were intercompared to ground-based microwave radiometer measurements at the NDACC sites Lauder, New Zealand and Mauna Loa, Hawaii (Boyd et al., 2007). The profiles agree within 5 % in the range of 18–0.04 hPa.

## 3.5 ACE-FTS

ACE-FTS is a high resolution infrared Fourier Transform Spectrometer and the main instrument of the Canadian Atmospheric Chemistry Experiment satellite mission which is also called SCISAT (Bernath, 2017). The satellite is in a 73.9° orbit at 650 km altitude. ACE-FTS performs solar occultation measurements from 2.2–13.3$\mu$m. The retrieved profiles have an altitude range of 5–95 km for ozone and 5–101 km for water vapour and an altitude resolution of 3–4 km. Within 3 years ACE measured only 6 profiles which were within $\pm 2°$ latitude and $\pm 4°$ longitude of Ny-Ålesund. For this study we use data from the retrieval version 3.6.

A global intercomparison of ACE-FTS ozone and water vapour profiles with MLS and MIPAS profiles revealed an average bias of -5 % between 20–60 km for water vapour. ACE-FTS ozone profiles are within $\pm 5$ % of MLS and MIPAS profiles in the mid stratosphere and exhibit a positive bias of 10–20 % in the upper stratosphere and lower mesosphere (Sheese et al., 2017).

## 3.6 SD-WACCM

The Whole Atmosphere Community Climate Model (WACCM, Marsh et al., 2013) is the "high-top" atmospheric component of NCAR's Community Earth System model (CESM). It is based on the Community Atmosphere Model (CAM, Collins et al. (2006)) and the chemistry module is taken from the Model for Ozone and Related Chemical Tracers (MOZART, Emmons et al. (2010)). For this study we use the specified dynamics version of WACCM (SD-WACCM, Brakebusch et al., 2013)) within CESM version 1.2.2. The model spans from ground to an altitude of 145 km with 88 levels and a vertical resolution of 0.5–4 km. It has a horizontal resolution of 1.9° latitude and 2.5° longitude and a time resolution of 30 minutes. In the specified dynamics version horizontal winds, temperature, surface wind stress, surface pressure and specific and latent heat flux are

nudged with GEOS5 analysis data (Rienecker et al., 2008). The nudging is performed at every time step with a strength of 10 % and up to an altitude of 60 km. Hoffmann et al. (2012) has shown that the nudging, although only 10 % and up to 60 km altitude, yields realistic variations of carbon monoxide (CO) in the higher mesosphere.

## 3.7 ERA5

ERA5 (Copernicus Climate Change Service (C3S), 2017; Hersbach and Dee, 2016) is a climate re-analysis of the European Center for Medium Range Weather Forecast (ECMWF). For its production a 4D Var data assimilation scheme was used in the Integrated Forecast System (IFS Cycle 41r2). For the considered time period ozone is assimilated from retrievals of the GOME-2 instruments on the METOP-A/B satellites, the SBUV-2 instruments on the NOAA satellites and of MLS and OMI on the EOS-Aura satellite up to 50 km. Water vapour is assimilated in the troposphere from humidity profiles measured with radiosondes and from ground stations which are provided by the World Meteorological Organizations Information System (WMO WIS). ERA5 provides hourly analysis fields at a horizontal resolution of 31 km and from the surface up to 0.01 hPa (about 80km). The vertical resolution is 20 m–2.5 km.

## 3.8 Ozone radiosonde

Balloon-borne ozone radiosondes are launched once per week at 11 UTC from the AWIPEV station at Ny-Ålesund. During polar night the launch frequency is enhanced to two sondes per week and during measurement campaigns even to one sonde per day. The used ozone sensor is an electrochemical concentration cell (ECC) model 6A. The weather balloons which carry the ozone sondes reaches altitudes of about 30 km within 1 h 40 min. A measurement is performed every 5 seconds which leads to an average vertical resolution of 30 m.

## 4 Methods

### 4.1 $H_2O$ descent rate

To determine the water vapour descent rate a linear least squares fit of the 5, 5.5 and 6 ppm isopleths were performed between September 15 and November 1 for the years 2015–2017. During these time intervals the mesosphere above Ny-Ålesund was always inside of the polar vortex system and the MIAWARA-C water vapour time series showed a linear descent. The 5.5 ppm isopleths of MIAWARA-C covered a large part of the mesosphere (0.02–0.4 hPa, approximately 73–53 km) within this time interval.

We assumed an uncertainty of 10 % for the MIAWARA-C water vapour VMR measurement and 5 % for the MLS measurements. The water vapour VMR in the mesosphere is linearly decreasing with altitude and we can determine the inverse of the average vertical water vapour gradient around the 5.5 ppm isopleth which is 6 km/ppm. Multiplied with the uncertainty in the measured VMR this leads to an altitude uncertainty of 3.3 km for MIAWARA-C and 1.6 km for MLS. The propagation of the altitude uncertainty in the least square fit leads then to the uncertainty of the slope of the fit.

## 4.2 Residual mean meridional circulation

The residual mean meridional circulation, also known as the Transformed Eulerian Mean (TEM) circulation (Andrews and McIntyre, 1976), describes the bulk motion of air parcels in the meridional and vertical direction. We calculate the vertical velocity of the TEM circulation (Smith et al., 2011):

$$\overline{w}^* = \overline{w} + \frac{1}{a\cos(\phi)}\frac{\partial}{\partial\phi}\left(\cos(\phi)\frac{\overline{v'\theta'}}{\partial\overline{\theta}/\partial z}\right), \tag{1}$$

where $\overline{w}$ is the zonal mean vertical wind and $v'$ is the perturbation component of the meridional wind. $a$ is the Earth's radius, $\phi$ the latitude, $\theta$ the potential temperature and $z$ the geopotential height. The fields to calculate $\overline{w}^*$ are taken from an SD-WACCM simulation. For comparison with the water vapour descent rates we calculated the mean velocity for the same altitude range and time period which was covered by the 5.5 ppm isopleth of MIAWARA-C or SD-WACCM and we calculated the mean velocity along the fit of the isopleth where we took the data point closest to the isopleth for every time step.

## 4.3 Polar vortex edge

For the discussion of the SSWs we determined the edge of the polar vortex from ECMWF geopotential height (GPH) and wind fields as the GPH contours with the highest wind speed at a given pressure level. The contour is calculated every 6 hours and at 50 levels between 18 and 75 km altitude. This definition for the polar vortex edge is used because it performs well from the stratosphere up to the mesosphere and even during an SSW. This is shown in Scheiben et al. (2012) where the method is also discussed in detail. Another possibility to define the polar vortex edge is to use the maximum of the potential vorticity gradients along potential vorticity isolines (Nash et al., 1996). Potential vorticity is an excellent tracer for the polar vortex edge in the stratosphere. It is however no longer a vortex centred coordinate above 60 km (Harvey et al., 2009) and can not be used to determine the polar vortex edge in the mesosphere.

For every model level and every 6-hour time step we calculated if Ny-Ålesund is in or outside of the polar vortex. The contour line of this inside/outside field indicates when the polar vortex edge was passing at Ny-Ålesund.

## 4.4 Lagranto backward trajectories

Backward trajectories are calculated with the lagrangian analysis tool (LAGRANTO, Sprenger and Wernli, 2015) using wind fields from the ECMWF operational data. The trajectories are started at Ny-Ålesund every 6 hours for 8 pressure levels between 100 and 0.03 hPa. With the trajectories we find the geographic origin of the air parcels arriving at Ny-Ålesund.

## 4.5 Bandpass filtering

Water vapour and ozone VMRs above Ny-Ålesund have been measured with a very high time resolution in the order of hours with our ground-based microwave radiometers. The time series are therefore well suited for an analysis of short term fluctuations in the Arctic middle atmosphere. For the spectral decomposition of the water vapour and ozone time series we chose a wavelet like approach. The time series were filtered with a digital non-recursive, zero-phase finite impulse response

filter using a Hamming window whose length is 3 times the centre period (Hocke and Kämpfer, 2008; Hocke, 2009). The advantage of a wavelet like approach is that it captures intermittent waves with non-persistent phase. The time series were separately filtered at every pressure level and for periods of 1–17 days.

## 5  $H_2O$ and $O_3$ data sets of MIAWARA-C and GROMOS-C

The ground-based microwave radiometers MIAWARA-C and GROMOS-C gathered a 3 year long and almost continuous time series of middle atmospheric water vapour and ozone volume mixing ratio (VMR) in the Arctic. The instruments are located at the AWIPEV research station at Ny-Ålesund, Svalbard (79°N, 12°E) and the measurements started in September 2015 and are ongoing.

### 5.1  $H_2O$ time series of MIAWARA-C

The 3 year long data set of water VMR measured with MIAWARA-C is presented in Fig.1. It shows a time series of water vapour VMR profiles in the altitude range of 10–0.005 hPa which corresponds approximately to 30–80 km.

The horizontal white lines indicate the upper and lower bound of the trustworthy altitude range. It is defined as the region where the measurement response is larger than 0.8. The measurement response for an altitude level is given by the area below the corresponding averaging kernel. The optimal estimation method, which is used to retrieve a profile out of the measured

spectra, needs additional information which is given in the form of an a priori profile. The measurement response is a measure for how large the contribution of the measurement is compared to the contribution of the a priori profile. A measurement response larger than 0.8 means that the measured spectrum contributes more than 80 % to the retrieved VMR. The trustworthy altitude range is larger in winter when the opacity of the troposphere is lower. The vertical grey lines are data gaps because of power cuts or because of rain since measurements during rain are not possible with MIAWARA-C.

The time series shows the annual cycle of water vapour VMR. In the mesosphere at 0.1 hPa water vapour has a maximum in summer of about 7.5 ppm and in winter it decreases to about 3.5 ppm. In the upper stratosphere at 5 hPa the maximum of about 7.5 ppm is seen in autumn when mesospheric water vapour is descending in the winter polar vortex and the minimum of about 6 ppm is seen in spring. The effective descent rate of water vapour during the formation of the polar vortex of the three winters since September 2015 is discussed in section 7. During winter the variability of water vapour is dominated by the dynamics of

the polar vortex. Horizontal water vapour VMR gradients across the vortex edge lead to variations in the water vapour mixing ratios above Ny-Ålesund when the vortex moves away from Ny-Ålesund. This is mainly seen during winter 2016/2017 and is discussed in section 8.4.

### 5.2  $O_3$ time series of GROMOS-C

Figure 2 presents the time series of ozone VMR measured with GROMOS-C over 3 years. The ozone profiles cover an altitude

range of 100–0.03 hPa which corresponds to about 15–70 km. The horizontal white lines indicate the measurement response

of 0.8, smoothed over 2 days. Data gaps are indicated with the vertical grey lines. During winter 2017/2018 GROMOS-C measured CO for about 2 months and during winter 2016/2017 the spectrometer had a hardware problem.

The main ozone layer at about 35 km is clearly seen as well as the annual cycle with higher ozone VMR in summer (6 ppm) than in winter (4.5 ppm). Stratospheric ozone VMR above Ny-Ålesund depends on the dynamics of the winter polar vortex.
This is seen from January to April 2017 (Sect. 8.4) as well as during two major sudden stratospheric warmings in March 2016 and February 2018 where stratospheric ozone VMR reached exceptionally high values of more than 8 ppm (Sect. 8). The measurements of the year 2016 have been used to study the diurnal ozone variations throughout the year (Schranz et al., 2018). In the mesosphere a diurnal cycle was detected in spring and autumn when there is light and darkness within one day. Ozone is depleted through photodissociation during daytime and subsequently recombines at night which leads to a diurnal ozone
variation of up to 1 ppm at 0.1 hPa. In the stratosphere the diurnal variations are seen throughout the polar day. At 10 hPa the largest variations of about 0.3 ppm are seen around summer solstice. At this altitude the net ozone production is positive for a solar zenith angle smaller than 65–75°, depending on the season, otherwise the net production is negative which leads to an ozone maximum in the late afternoon. At 1 hPa around the stratopause the diurnal cycle has the largest amplitudes of 0.5 ppm in the end of April/beginning of May and in August. The chemistry of diurnal ozone variations in general is described in
Schanz et al. (2014) and specifically for Ny-Ålesund in Schranz et al. (2018).

## 6  Intercomparison

The water vapour and ozone datasets of the two ground-based microwave radiometers of the University of Bern were intercompared with satellite measurements of MLS and ACE and with the model SD-WACCM and the reanalysis ERA5. The ozone time series was additionally intercompared with balloon borne ozone measurements where there was a reasonable overlap in
altitude and measurements from the ground-based microwave radiometer OZORAM. The left panels in the Figs. 3 and 5 show the daily mean VMR of all the different datasets averaged within four pressure intervals covering 3–0.03 hPa for water vapour and five pressure intervals covering 30–0.1 hPa for ozone. The time series start in September 2015 and end in September 2018. In the right panel the relative differences of the different datasets to MIAWARA-C and GROMOS-C measurements are shown.

An intercomparison of the profiles was also performed. The time series was divided into bins according to the integration
time of MIAWARA-C (6 hours) and GROMOS-C (2 hours). The profiles of the other instruments or models which fall into a bin were then averaged and convolved with the averaging kernels of the instruments and the relative difference between the profiles was calculated. The OZORAM data were not convolved because the vertical resolution is comparable to the one of GROMOS-C. For the balloon borne ozone sonde measurements the relative difference for both the convolved and unconvolved data are shown because a meaningful convolution is only possible up to 20 hPa. In Figs. 4 and 6 the median of the relative
difference profiles is shown (left) as well as their median absolute deviation from the median (MAD) which is defined as $MAD = \text{median}_i |x_i - \bar{x}|$ where $\bar{x} = \text{median}_j(x_j)$ as measure for the spread (right).

## 6.1 H₂O intercomparison

The intercomparison of the MIAWARA-C water vapour time series with measurements of the satellite instruments MLS and ACE as well as with the model SD-WACCM and the reanalysis ERA5 shows that MIAWARA-C and SD-WACCM agree within 10 % in the lowest panel (1-3 hPa) and that the six ACE measurements are even within 5 % (Fig. 3). ERA5 shows less water vapour in late summer but is also mostly within ±10 % of MIAWARA-C. MLS measurements, however, have an offset of about 15 % which is constant over the whole time period and it persists also at the higher altitude levels. With increasing altitude SD-WACCM and ERA5 start to see less water vapour in winter and more water vapour in early summer than MIAWARA-C. ACE stays close to MIAWARA-C also at higher altitudes, it is however always higher than MIAWARA-C and the ACE measurements cover only the period of end of September until mid October.

The median of the relative difference profiles (Fig. 4) shows that SD-WACCM and ACE are within ±5 % of MIAWARA-C up to 0.1 hPa. MLS has a median offset of 10–15 % whereas for ERA5 the median relative difference is -3– -13 %. Above 0.1 hPa MIAWARA-C starts to see water vapour than MLS, SD-WACCM and ACE.

An offset of MLS (13 %) in the mesosphere was already seen in an earlier intercomparison study of MIAWARA-C at Bern and Sodankylä. In the SPARC water vapour assessment for the years 2004–2015 the offset of MLS to ground-based microwave radiometers is also mentioned (Nedoluha et al., 2017) but the mean relative difference (0–10 %) is smaller than for MIAWARA-C. At Thule, Greenland Mevi et al. (2018) measured water vapour with a ground-based microwave radiometer for 1 year starting in July 2016 and noticed no clear difference to MLS.

## 6.2 O₃ intercomparison

The ozone time series measured with GROMOS-C is intercompared with satellite data sets of MLS and ACE, with the model SD-WACCM and with the reanalysis ERA5 in Fig. 5. In the lowest pressure interval (30–10 hPa) the GROMOS-C measurements are also intercompared to balloon-borne ozone sonde measurements and from 10–0.1 hPa they are additionally compared to the OZORAM measurements. In the lowest panel (30–10 hPa) all models and instruments agree with each other except for GROMOS-C which measures up to 20 % higher ozone VMRs in summer whereas in winter it is mainly within 10 % of the other datasets. This annual variation of the relative differences to GROMOS-C persists up to 1 hPa. At 1–0.3 hPa GROMOS-C agrees with OZORAM well within 10 % whereas SD-WACCM and MLS show lower VMRs. At 0.3–0.1 hPa the relative differences are high due to very low VMRs but the datasets agree well with each other. Up to 3 hPa ERA5 ozone VMR agrees well with the other datasets but above it starts to deviate, mainly during summer but also in winter at 0.3–0.1 hPa. This is because ERA5 uses an ozone parametrization (Cariolle and Teyss, 2007) which is designed for the stratosphere and because there is no ozone assimilation in the mesosphere. The relative difference of ERA5 to GROMOS-C at 0.3–0.1 hPa ranges from 100 % in summer to -50 % in winter.

Up to 0.5 hPa (about 55 km) the median of the differences relative to GROMOS-C is mainly within 5 % for OZORAM (above 6 hPa), MLS, ACE, SD-WACCM and ERA5 (Fig. 6). The relative difference of the balloon borne ozone sonde to

GROMOS-C is increasing from -3 % at 30 hPa to -13 % at 10 hPa. In general the ozone measurements of GROMOS-C are up to 5 % higher than OZORAM, MLS, SD-WACCM and ERA5.

## 7    Effective descent rate of $H_2O$

Middle atmospheric air at the poles is descending within the polar vortex during its formation in autumn. For the autumns 2015–2017 we determined the effective descent rate of water vapour in the mesosphere above Ny-Ålesund from the MIAWARA-C and MLS measurements, the SD-WACCM simulation and the ERA5 reanalysis. For MLS and SD-WACCM we also determined the zonal mean descent rate at the latitude of 79° N. These descent rates were then compared to the mean residual circulation calculated from SD-WACCM data according to Smith et al. (2011). The methods for calculating the water vapour descent rate and the mean residual circulation are described in Sec. 4.

From the 5.5 ppm contour of the MIAWARA-C water vapour time series we got effective descent rates of 428±12 m/day, 404±12 m/day and 468±13 m/day (Fig. 7) between September 15. and November 1. of the years 2015, 2016 and 2017. The descent rates were also calculated for different isopleths. At 5 ppm the descent rates are within ±4 % of the results from the 5.5 ppm isopleth and at 6 ppm the descent rates are 2–12.5 % lower. In Fig. 8 the water vapour descent rates from MIAWARA-C are compared to the descent rates of MLS, SD-WACCM and ERA5. The solid line connects descent rates from the 5.5 ppm isopleth at Ny-Ålesund and the dashed line indicates zonal mean descent rates. Descent rates from the 5 and 6 ppm isopleths are indicated with different symbols. The descent rates from the MLS water vapour measurements at Ny-Ålesund and for the zonal mean are within 10 % of MIAWARA-C. The model and the ranalysis do however have an average discrepancy of +30 % for SD-WACCM and -20 % for ERA5 at Ny-Ålesund. The average discrepancy for the zonal mean of SD-WACCM is 45 %. Figure 9 shows mean profiles of the vertical component of the mean meridional circulation $\overline{w}^*$ and the vertical wind from SD-WACCM for September 1 until November 1 of the years 2015, 2016 and 2017. The maximum of $\overline{w}^*$ is at around 0.07 hPa with 1000–1350 m/day, towards 1 hPa it decreases to 350–400 m/day. The bold line in Fig. 9 indicates the altitude range covered by the 5.5 ppm zonal mean water vapour isopleth of SD-WACCM and the points indicate the mean over these altitude ranges. The averaged $\overline{w}^*$ range from 800 to 910 m/day which is higher than the water vapour descent rate from SD-WACCM. If we average $\overline{w}^*$ along the fit of the zonal mean 5.5 ppm isopleths of SD-WACCM the velocities decrease in the year 2015 and slightly increase in the years 2016 and 2017 and differ by 16 %, 39 % and 38 % from the zonal mean water vapour descent rates of SD-WACCM. The averaged zonal mean vertical wind profiles show a higher descent rate than $\overline{w}^*$ in the upper mesosphere and a smaller descent rate in the lower mesosphere.

The descent rates of trace species in the Arctic autumn have been estimated previously. In the mesosphere at 75 km Forkman et al. (2005) found a descent rate of up to 300 m/day from CO and $H_2O$ measurements with ground-based microwave radiometers located at 60°N. Funke et al. (2009) found CO descent rates of 350-400 m/day at 50–70 km in September and October averaged for 60–90° N from MIPAS satellite measurements. The average descent rate from October to February in the lower mesosphere was determined from ACE-FTS $CH_4$ and $H_2O$ measurements and is 175 m/day (Nassar et al., 2005).

Ryan et al. (2018) were assessing the ability to derive middle atmospheric descent rates from trace gas measurements. For CO they concluded that CO chemistry and dynamical processes other than vertical advection are not negligible and that the CO descent rate does not represent the mean descent of the atmosphere. Therefore Ryan et al. (2018) suggested to interpret the descent rate of trace species as an effective rate of vertical transport of this trace species. Water vapour has a longer lifetime than CO at 50–70 km (Brasseur and Solomon, 2005) which makes it a more robust tracer. The SD-WACCM simulations show that $\overline{w}^*$, if it is averaged along the 5.5 ppm water vapour isopleth of SD-WACCM, is within 16–39 % of the zonal mean water vapour descent rates. This shows that in general water vapour is a rough proxy for the vertical bulk motion in the high Arctic during the formation of the polar vortex in autumn. The difference of 16–39 % shows that other processes than vertical advection contribute to the effective descent rate of water vapour in the polar vortex.

The large difference between the mesospheric water vapour descent rate from SD-WACCM and ERA5 seems to indicate that the model and the ranalysis have difficulties to catch the autumn descent of water vapour at high latitudes. It is also seen in the increasing relative differences between the model and reanalysis and the measurements in autumn and spring (see Fig. 3) when air parcels are descending and rising respectively. In an intercomparison of SD-WACCM CO data with measurements from MLS and the ground based microwave radiometer KIMRA at Kiruna (68° N) SD-WACCM shows higher CO VMRs in autumn and spring at high altitudes (86 and 76 km) (Ryan et al., 2018). The CO VMR is increasing with altitude whereas the $H_2O$ VMR is decreasing. Therefore too high CO and too low $H_2O$ VMRs in autumn could indicate a too strong mesospheric descent within SD-WACCM. But also difficulties with the $H_2O$ and CO chemistry might play a role because in spring SD-WACCM CO and $H_2O$ are both higher than the measurements.

## 8 Major and minor sudden stratospheric warmings of 2016–2018

In the years 2015–2018 two major sudden stratospheric warmings (SSW) with a split of the polar vortex and several minor warmings took place. For the characterisation of the event as minor or major warming we follow the definition of the World Meteorological Organisation (WMO) as presented in McInturff (1978): A stratospheric warming is called minor if a significant temperature increase is observed (i.e., at least 25 K in a period of a week or less) at any stratospheric level in any area of the wintertime hemisphere and if the criteria for major warmings are not met. A stratospheric warming is major if at 10 hPa or below the latitudinal mean temperature increases poleward from 60° latitude and an associated circulation reversal is observed (i.e., mean eastward winds poleward of 60° latitude are succeeded by mean westward winds in the same area).

In the following sections we present an analysis of the two major SSWs of the years 2016 and 2018 and the minor warmings of 2017 as seen from the perspective of Ny-Ålesund.

### 8.1 Major SSW of March 2016

The first major SSW which we observed at Ny-Ålesund took place in March 2016. Figure 10 shows the contour lines of the polar vortex at 10, 1 and 0.1 hPa (solid line) and the 3-day backward trajectories calculated with LAGRANTO. In the stratosphere at 10 hPa the polar vortex was elongated and shifted away from Ny-Ålesund on March 2 for the first time. It

regained a circular shape on March 5 before it shifted away from the pole again on March 9 and split on March 15. In the mesosphere (1 and 0.1 hPa) the vortex was shifted away from the pole too and it got the form of a horseshoe. The backward trajectories show that the air parcels arriving at Ny-Ålesund mainly followed the contours of the polar vortex and brought midlatitude air to the pole when the vortex was shifted. Because the polar vortex did not reestablish after the SSW and the

circulation directly went over to the summer anticyclone, the event is called a major stratospheric final warming (Manney and Lawrence, 2016).

Figure 11 shows temperature, zonal wind, water vapour and ozone during the SSW 2016 at Ny-Ålesund. The black lines indicate the dates in Fig. 10. At 10 hPa the warming started on March 4 and the temperature increased 60 K in 2 days. During the stratospheric warming the isentropes descended in the stratosphere which contributed to the observed temperature increase

through adiabatic heating of the descending airmasses. In the mesosphere the isentropes were rising and the upwelling of air along the isentropes led to adiabatic cooling and a temperature decrease of 40 K was measured at 0.01 hPa. The wind profiles from GROMOS-C captured the reversal of the zonal wind from eastward to westward in the mesosphere on March 5. In the stratosphere the zonal wind was already westward during February because at the latitude of Ny-Ålesund a slight shift of the polar vortex off the pole and towards Ny-Ålesund is enough to reverse the zonal wind.

When the polar vortex moved away from Ny-Ålesund on March 2 stratospheric ozone increased by 1.5 ppm because ozone richer air from the midlatitudes reached the polar region. In the water vapour time series an increase in the mesosphere and a decrease in the stratosphere was seen which corresponds to the vertical water vapour structure outside of the polar vortex.

## 8.2   Major SSW of February 2018

The second major stratospheric warming which we observed took place in February 2018. Figure 12 shows the contour of the

polar vortex at different stages during the SSW. In the stratosphere the polar vortex was elongated on February 9 and then it split in two on February 11 and moved away from Ny-Ålesund. On February 13 also the mesospheric vortex moved off the pole and on February 26 the mesospheric vortex reestablished whereas in the stratosphere the vortex was still split. From the trajectories we see that the airmasses arriving at Ny-Ålesund moved along the polar vortex contour except when the vortex split on February 12 the air arrived straight from the Atlantic in the stratosphere and from central Europe in the mesosphere.

Figure 13 shows temperature, zonal wind, water vapour and ozone during the SSW 2018 at Ny-Ålesund and the black lines indicate the dates in Fig. 12. The stratospheric temperature between 100 and 10 hPa increased almost at the same time by 35 K or more within 3 days. In the mesosphere the temperature decrease started already on February 6. The rise of the isentropes in the mesosphere and the descent in the stratosphere indicate a corresponding motion of airmasses which is actually found in the vertical velocity $\overline{w}^*$ of the mean residual circulation. With the sudden stratospheric warming the temperature distribution

in the stratosphere and mesosphere was almost homogeneous in the whole altitude range and the stratopause was no longer clearly defined.

The zonal wind measurements show a lot of changes in direction as it depends on the position of the polar vortex. Ozone increased dramatically by 4 ppm reaching a VMR of 8 ppm in the stratosphere when the vortex split on February 10 and midlatitude air from the Atlantic was brought to Ny-Ålesund. Ongoing meridional mixing brought again ozone rich air from

the centre of the United States on February 25 and the ozone VMR increased again to 8 ppm. In the water vapour time series a VMR increase was already seen on February 1 which is followed by a descent similar to the water vapour descent observed during the formation of the polar vortex. The mesospheric water vapour increase was accompanied by a water vapour decrease in the upper stratosphere.

## 8.3  Ozone and water vapour during the major SSWs

During the two major SSWs in 2016 and 2018 stratospheric ozone and mesospheric water vapour were enhanced at Ny-Ålesund. The ozone enhancement is in agreement with the results of a composite analysis of 20 major SSWs in the ERA interim reanalysis dataset who found an increase in the total ozone column in the Arctic region after an SSW (Hocke et al., 2015). In contrast to the Arctic, at midlatitudes observations of stratospheric ozone during the 2008 SSW showed an ozone depletion along with the temperature increase (Flury et al., 2009). The depletion was mainly attributed to the effects of higher temperatures on the ozone chemistry especially to a higher efficiency of the catalytic ozone destruction through NOx. At Ny-Ålesund the net chemical ozone production rate in the stratosphere, taken from SD-WACCM, is always negative during the periods when the 2 SSWs took place and the ozone enhancements result therefore solely from transport of ozone rich midlatitude air to the pole. The ozone increase of February 25 is therefore not a result of the decrease in temperature which was observed at the same time.

The midlatitude air also brought moister air to the mesosphere at Ny-Ålesund whereas in the stratosphere (around 3 hPa) water vapour decreased. Midlatitude air is drier than vortex air at that altitude because of the subsidence of water vapour rich airmasses from higher stratospheric levels inside of the polar vortex (Lossow et al., 2009). The evolution of water vapour during SSWs has been observed from midlatitude sites in earlier studies. At sites in Europe an increase in mesospheric water vapour was measured during the SSWs in the years 2008, 2010, 2012 and 2013 (Flury et al., 2009; Straub et al., 2012; Tschanz and Kämpfer, 2015) whereas in South Korea water vapour was decreasing during the 2008 SSW (De Wachter et al., 2011). The descent rate after the 2018 SSW in the lower mesosphere was calculated as explained in Sec. 7 for the period March 1–15 and is 355 m/day.

This is in agreement with estimates of the water vapour descent rates from Sodankylä ($67°$ N) which are 350 m/day in 2010, 364 m/day in 2012 and 315 m/day in 2013 (Straub et al., 2012; Tschanz and Kämpfer, 2015). From water vapour and methane measurements with SOFIE Bailey et al. (2014) find a descent rate of 345 m/days after the 2013 SSW at approximately $70°$ N and between 50 and 60 km altitude.

Straub et al. (2012) showed that descent rates estimated with MIAWARA-C water vapour time series at Sodankylä after an SSW are in agreement with Transformed Eulerian Mean (TEM) trajectories derived from SD-WACCM simulations and that therefore the effective descent rates of water vapour are an estimate for the atmospheric residual circulation. We found that the mean residual circulation averaged along the fit of the 5.5 ppm isopleth of MIAWARA-C from March 1–15 has a vertical velocity of 473 m/day which is 33 % higher than our estimate. The zonal mean water vapour descent rate from SD-WACCM however agrees within 10 % with $\overline{w}^*$ when it is averaged along the fit of the 5.5 ppm water vapour isopleth of SD-WACCM. This is in contrast to Ryan et al. (2018) which assessed the ability to derive atmospheric descent rates from CO and found that

in general after an SSW other processes affect the CO VMR more than vertical advection. However in the region between 67 and 57 km altitude and from about 10 to 40 days after the SSW 2009 the CO descent rate agrees to the vertical advection (Ryan et al., 2018, Fig. 8). This is the area where we and Straub et al. (2012) fitted the water vapour isopleths and it could explain the agreement of the water vapour descent rate from the isopleths with the mean residual circulation.

5    3-day backward trajectories were calculated with the lagrangian analysis tool LAGRANTO. They showed a high variability in the latitudinal origin of the airmasses at Ny-Ålesund during the 2 major SSWs. After the 2018 SSW the mesospheric polar vortex recovered in the end of February which is seen in the latitudinal origin returning to values larger than 60°latitude (not shown).

## 8.4    Minor warmings in winter 2017

10    From January to April 2017 water vapour showed several spikes in the mesosphere (see Fig. 1). Thereby the water vapour VMR in the mesosphere at 0.1 hPa is enhanced by 2 ppm for about 4–11 days. In Fig. 14 (top) the water vapour time series is shown and the black contour lines indicates when the polar vortex edge is right above Ny-Ålesund. It is evident that the enhancements in mesospheric water vapour coincide with the periods where the polar vortex is shifted away from Ny-Ålesund. In the ozone time series the shift of the vortex away from Ny-Ålesund is seen as an enhancement in stratospheric ozone of 15    about 2.5 ppm in January (Fig. 14 middle). In March the ozone increases in the middle stratosphere are less clearly linked to these shifts as during the period of polar night. From the zonal mean time series of ECMWF temperature and zonal wind we find that the first three shifts meet the criteria of a minor sudden stratospheric warming according to the definition mentioned above.

During the major sudden stratospheric warmings of 2016 and 2018 (Sec. 8.1 and 8.2) the polar vortex system completely 20    broke down and mixing of the airmasses occurred. The strong water vapour and ozone changes which accompanied the vortex shifts indicate that, in contrast to the major SSWs, the polar airmasses stayed clearly separated from midlatitude air at the polar vortex edge. An analysis of LAGRANTO 3-day backward trajectories shows the latitudinal origin of the airmasses at Ny-Ålesund. During the shifts of the upper stratospheric and mesospheric vortex the airmasses arrive from the midlatitudes (Fig.14 bottom) which confirms that the separation of polar and midlatitude air persists during the minor warmings.

## 25  9    Periodicities

Planetary waves are global-scale, coherent perturbations in the atmospheric circulation. Interactions of the waves with the mean-flow influence the large-scale dynamics (Andrews et al., 1987) and are believed to be the main cause of SSWs (Matsuno, 1970). The dominant periods of planetary waves were found at about 2, 5, 10 and 16 days in datasets from ground based instruments and from satellites (e.g. Lainer et al. (2018); Pancheva et al. (2018); Tschanz and Kämpfer (2015); Forbes and 30    Zhang (2015); Scheiben et al. (2014); Day and Mitchell (2010); Riggin et al. (2006)). These periods correspond to the numerically calculated periods of Rossby normal modes and Rossby-gravity waves (Salby, 1981). Because of the variability in their

wave periods the waves are called quasi-2-day wave (Q2DW), quasi-5-day wave (Q5DW), quasi-10-day wave (Q10DW), and quasi-16-day wave (Q16DW).

The water vapour and ozone time series measured with MIAWARA-C and GROMOS-C are almost continuous and have a high time resolution. We applied a bandpass filter with periods of 1–17 days to investigate the periodicities in the time series. Figure 15 presents an overview of the wave activity in the water vapour and ozone time series during summer and winter. It shows the mean peak-to-peak amplitude of the filtered signal for the different periods and pressure levels. The amplitudes are averaged over 80 day intervals centred at summer and winter solstices of the years 2015–2018. For ozone the figure is only shown for summer because in winter the ozone time series had long gaps.

Remarkable is the summer 2016 where the water vapour time series shows high amplitudes for periods of about 2, 5 and 10 days in the mesosphere at 0.1–0.01 hPa and for a period of 16 days in the upper stratosphere which corresponds to the periods of all the dominant Rossby normal modes and the Rossby-gravity wave. Also in the following two summers the water vapour time series shows the largest amplitudes in the upper mesosphere. The periods where we see the largest amplitudes do however not so clearly correspond to the normal modes. In winter periodicities are found between 1 and 0.1 hPa and cover a larger altitude range than in summer. In the winter 2016/2017 a strong Q5DW is present which is not seen in the other years. A persistent feature in winter is a wave with a period of about 3 days which is not attributed to the quasi 2- or 5-day wave. In general we note a high interannual variability in the dominant periods of the waves whereas the general pattern of the wave periods is repeated. Below 10 hPa no periodicities are seen in the water vapour measurements of MIAWARA-C.

In the ozone time series periodicities are seen in the stratosphere and the amplitudes are lower than for water vapour. The ozone time series in summer show signatures of a Q16DW with periods of 12–19 days in the stratosphere at about 10 hPa. In summer 2016 a Q16DW is found at the same altitude as for water vapour.

Additionally a comparison of the wave patterns in the water vapour and ozone time series from MIAWARA-C and GROMOS-C with SD-WACCM is shown in Figs. 16 and 17. SD-WACCM water vapour shows only small amplitudes (<0.05 ppm) during summer, therefore we compared the amplitudes for winter and in the mesosphere whereas for ozone we look at the summer stratosphere. The periods were averaged over the 80 day periods centred around winter or summer solstice of the years 2015-2018. For water vapour we find that SD-WACCM and MIAWARA-C show a similar distribution of the wave amplitudes for the lower mesosphere (0.1–1 hPa) whereas in the upper mesosphere (0.01–0.1 hPa) there is no agreement between SD-WACCM and MIAWARA-C. In the ozone time series at 5–50 hPa the diurnal cycle has a large amplitude. SD-WACCM shows a stronger drop at a period of two days but in general both datasets show increasing wave amplitudes for growing period lengths. The peaks at 8 and 15 days are however not captured by SD-WACCM.

Periodicities in the water vapour time series are mainly seen at 0.1–0.01 hPa during summer whereas in winter periodicities are seen throughout the mesosphere (Fig. 15). The seasonal differences in the altitude distribution of the periodicities agree with the findings of Tschanz and Kämpfer (2015) and Pancheva et al. (2018). In water vapour and geopotential height data they found that the amplitudes of Q2DWs peak at higher altitudes in summer than in winter. Additionally Day et al. (2011) found from temperature data that the Q16DW is present throughout the mesosphere in winter whereas in summer it is present only

below 30 km and above 70 km. This pattern can also be recognised in our water vapour dataset where we see very low wave activity around the stratopause (1–0.1 hPa) in summer for periods around 16 days and high activity in winter.

The chemical lifetime of ozone in the summer stratosphere is about 20-40 minutes according to SD-WACCM which is shorter than the time scale of transport. Nevertheless periodicities are found in the ozone time series. They might result from temperature variabilities because of temperature dependent photochemistry (Moreira et al., 2016; Flury et al., 2009; Pendlebury et al., 2008). We find that the SD-WACCM temperature time series show the same amplitude pattern as SD-WACCM ozone at Ny-Ålesund. For both datasets the amplitudes of the periodicities are increasing with growing period lengths.

## 10    Summary and Conclusions

Continuous observations of water vapour and ozone in the Arctic middle atmosphere are rare. We presented a 3-year long time series of unique water vapour and ozone measurements in the Arctic middle atmosphere. These datasets have been measured with the two ground-based microwave radiometers MIAWARA-C and GROMOS-C at the AWIPEV research base at Ny-Ålesund, Svalbard in the years 2015–2018. The datasets are almost continuous and characterized by a high time resolution in the order of hours which allows us to analyse phenomena on a wide range of time scales.

The water vapour and ozone datasets were intercompared with measurements from the satellite instruments AuraMLS and ACE-FTS and with the model SD-WACCM and and the reanalysis ERA5. Ozone data were additionally intercompared with measurements of the ozone radiometer OZORAM which is co-located at Ny-Ålesund and with balloon borne ozone sonde measurements when there was a reasonable overlap in altitude. On average SD-WACCM and ACE are within ±5 % of the MIAWARA-C water vapour measurements up to 0.1 hPa (about 60 km). The MLS measurements have however a constant offset to MIAWARA-C over the 3 years which is on average 10–15 % depending on altitude. In the mesosphere this offset was already seen when MIAWARA-C was located at Bern and Sodankylä for 2012-2013 (Tschanz et al., 2013). In the stratosphere GROMOS-C shows good agreement with the other datasets during winter whereas in summer GROMOS-C measures up to 20 % higher ozone values than the other datasets. On average GROMOS-C profiles are mainly within 5 % of the other datasets up to 0.5 hPa (about 55 km).

During the 3 years of observation at Ny-Ålesund we measured water vapour, ozone and horizontal wind and observed dynamical phenomena which are typical for the Arctic middle atmosphere. The descent rate of mesospheric water vapour within the polar vortex in autumn from September 15 until November 1 was determined in the years 2015–2017 at 0.3–0.02 hPa. The effective descent rate determined from MIAWARA-C water vapour is 428, 404 and 468 m/day for 2015–2017. The descent rate determined with MLS water vapour measurements at Ny-Ålesund and for the zonal mean is within 10 % of the MIAWARA-C descent rate. The ERA5 reanalysis shows an average discrepancy of -20 % whereas the model SD-WACCM shows a discrepancy of 30 % at Ny-Ålesund and 45 % for the zonal mean. This reflects the deviations of model and reanalysis from the measurement in autumn. From the SD-WACCM data we calculated the vertical velocity $\overline{w}^*$ of the residual mean meridional circulation and find that $\overline{w}^*$ is within 16–39 % of the zonal mean water vapour descent rate of SD-WACCM. We conclude that the effective water vapour descent rate is in general a rough proxy for the vertical bulk motion at high latitudes

during the formation of the polar vortex in autumn. For a detailed description of the vertical motion from tracer descent rates other processes than vertical advection like horizontal advection and eddy transport need to be taken into account.

Two major sudden stratospheric warmings and a few minor warmings took place during the 3 years of observation. The major warmings took place in March 2016 and February 2018 whereas from January–April 2017 3 minor warmings took place. Enhancements in mesospheric water vapour and stratospheric ozone were observed and the wind reversals were captured in the GROMOS-C wind measurements. During the SSW in 2018 the stratospheric ozone VMR doubled and reached 8 ppm. This is in contrast to the midlatitudes where the temperature increases contribute to an ozone decrease. From SD-WACCM simulations we see that the stratospheric ozone enhancement is not linked to the temperature changes and that it is purely a transport effect. In contrast to the vortex shifts in 2017, where the polar airmasses were clearly separated from midlatitude air at the polar vortex edge, the polar vortex system broke down during the major SSWs 2016 and 2018 and mixing of the airmasses occurred. After the 2018 SSW the polar vortex reestablished and we determined the descent rate of water vapour which is 355 m/day for March 1–15. $\overline{w}^*$ is 33 % higher but with the zonal mean water vapour descent rate from SD-WACCM it agrees within 10 %. This is in contrast to the autumn period where $\overline{w}^*$ was 16–39 % higher than the zonal mean descent rate from SD-WACCM.

We presented an overview of the periodicities found in the water vapour and ozone time series. The water vapour and ozone time series were bandpass filtered and signatures of Rossby normal modes and Rossby-gravity waves with periods of 2, 5, 10 and 16 days were found. In the water vapour time series we note an interannual variation of the dominant periods whereas the general patterns of the wave periods are repeated. Seasonal differences from summer to winter for water vapour are seen in the peak altitude of the waves. In winter the largest amplitudes are mainly found around 0.1 hPa whereas in summer they are found around 0.03 hPa. A comparison with SD-WACCM shows that for water vapour in winter the periodicities are well captured at 0.1–1 hPa whereas at higher altitudes (0.01–0.1 hPa) SD-WACCM show up to 2 times larger amplitudes. In summer SD-WACCM wave amplitudes are very small (<0.05 ppm) whereas in the MIAWARA-C data signatures of the normal modes are found. For ozone in the summer stratosphere we found increasing amplitudes with growing period lengths for GROMOS-C and SD-WACCM. The same pattern was also seen in the temperature data from SD-WACCM which suggests a link between the periodicities in ozone and temperature.

We will continue the monitoring of ozone and water vapour in the variable polar middle atmosphere above Ny-Ålesund.

*Data availability.* The water vapour and ozone measurements from the NDACC instruments MIAWARA-C, GROMOS-C, OZORAM and the balloon borne ozone sonde are available at the NDACC data repository ftp://ftp.cpc.ncep.noaa.gov/ndacc/station/nyalsund/.

*Author contributions.* FS was responsible for the ground-based ozone measurements with GROMOS-C, performed the data analysis and prepared the manuscript. BT was responsible for the water vapour measurements with MIAWARA-C and MP for the ozone measurements with OZORAM. RR performed the wind retrieval from the GROMOS-C ozone spectra. KH designed the filter algorithm and NK contributed to the interpretation of the results.

*Competing interests.* The authors declare that they have no conflict of interest.

*Acknowledgements.* Observations with MIAWARA-C and GROMOS-C in Ny-Ålesund are funded by the Swiss National Science Foundation under grant number 200020-160048. For partial funding of this work we acknowledge the BMBF Germany (project 01LG1214A) and German Research Foundation (DFG) SFB/TR 172 Arctic Amplification: Climate Relevant Atmospheric and Surface Processes, and Feedback Mechanisms (AC)3 in projects B06 and E02. We thank the electronics workshop of the IAP and the AWIPEV team for their support during the campaign. In addition, we thank the satellite teams for providing the Aura/MLS and ACE/FTS data. The Atmospheric Chemistry Experiment (ACE), also known as SCISAT, is a Canadian-led mission mainly supported by the Canadian Space Agency. Finally we thank the two referees for their comments which helped to improve this article.

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

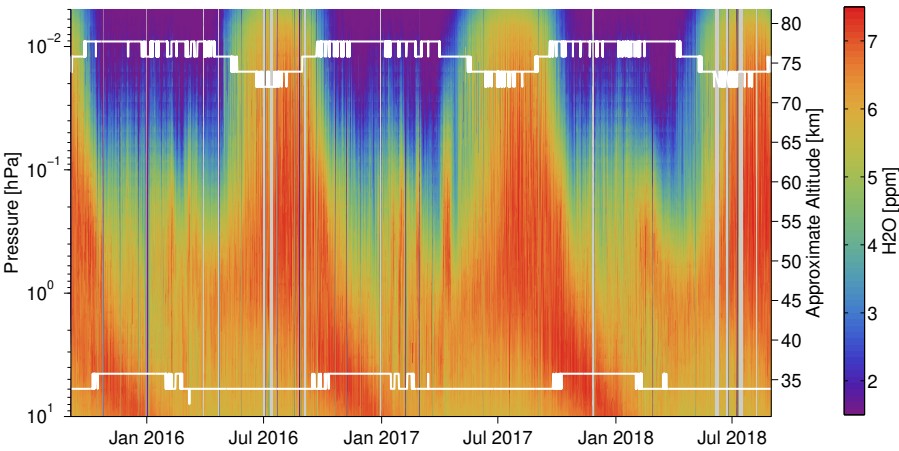

**Figure 1.** Time series of MIAWARA-C water vapour profiles from Ny-Ålesund. The horizontal white lines indicate the measurement response of 0.8.

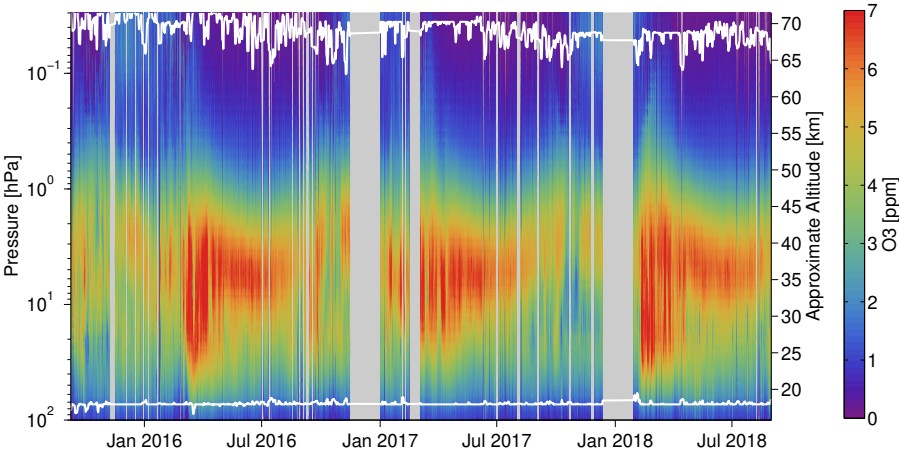

**Figure 2.** Time series of GROMOS-C ozone profiles from Ny-Ålesund. The horizontal white lines indicate the measurement response of 0.8.

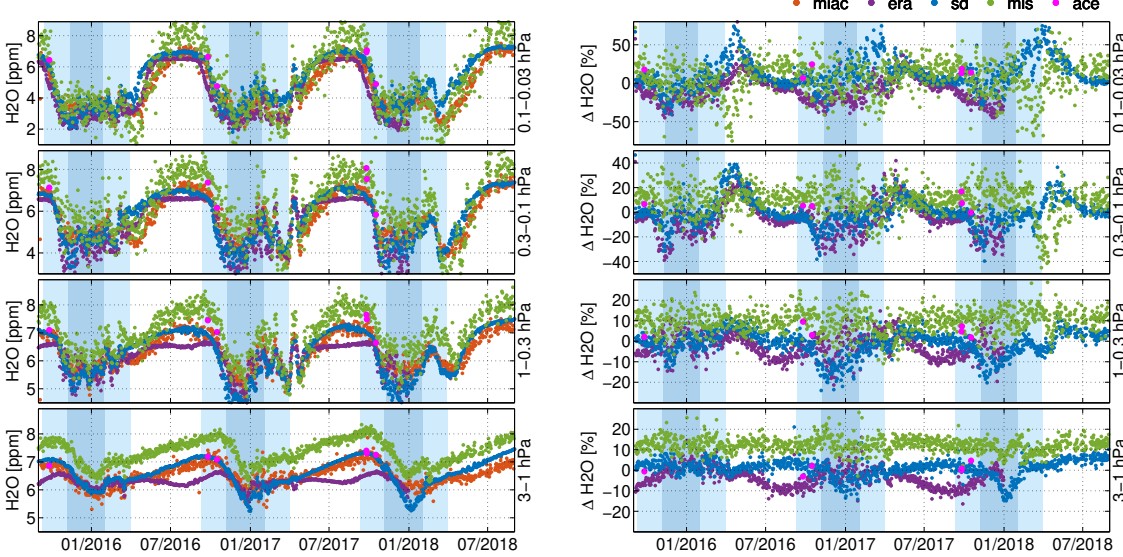

**Figure 3.** Intercomparison of water vapour time series at Ny-Ålesund. On the left are water vapour VMR time series of MIAWARA-C, MLS, ACE-FTS, SD-WACCM and ERA5 averaged within 4 pressure intervals where the upper 3 intervals are in the mesosphere and the lowest interval is in the upper stratosphere. On the right the relative differences to MIAWARA-C are shown for the same pressure intervals. The dark blue background indicates polar night and the white background polar day.

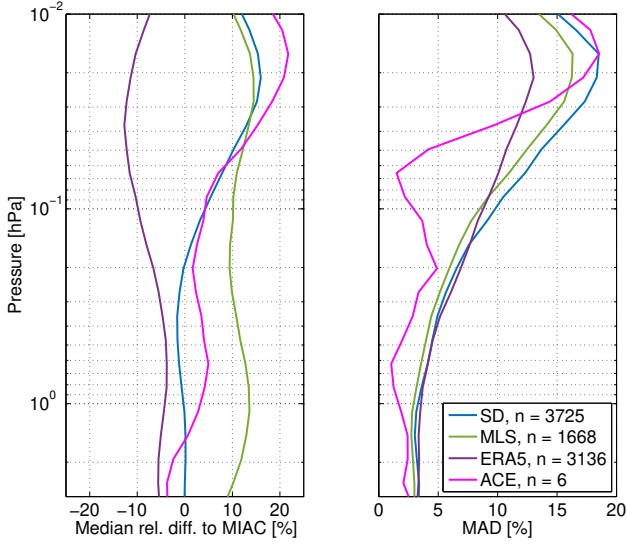

**Figure 4.** Median of the relative difference of SD-WACCM, MLS, ERA5 and ACE water vapour profiles and MIAWARA-C measurements at Ny-Ålesund (left). Median absolute deviation of the relative difference profiles (right). In the legend $n$ indicates the number of coincident MIAWARA-C profiles.

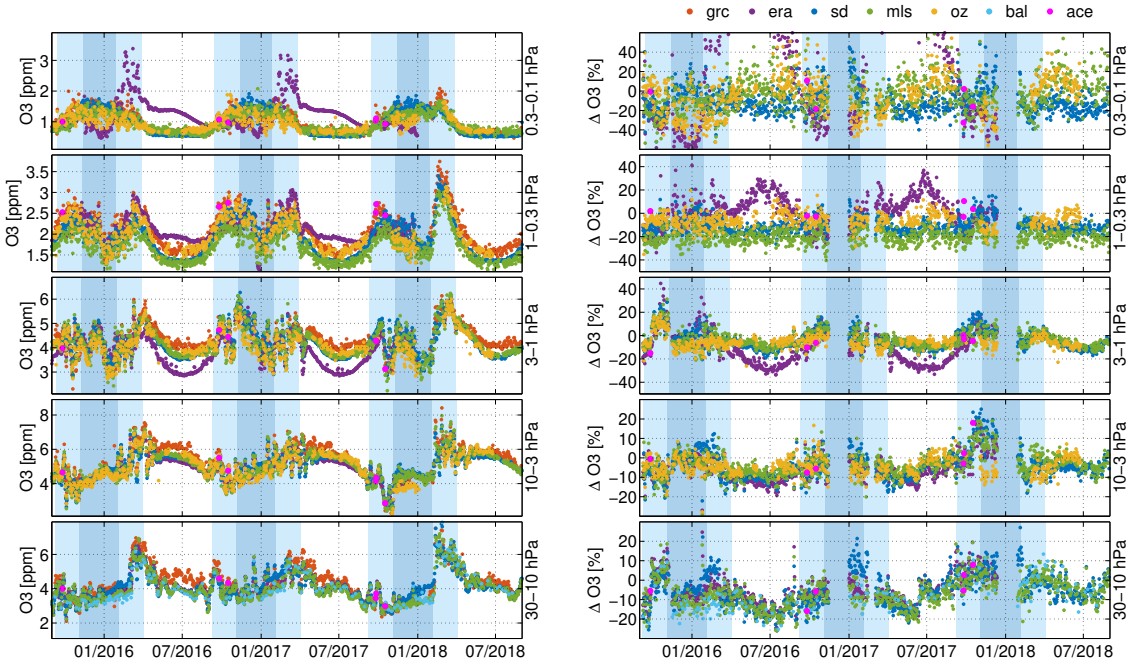

**Figure 5.** Intercomparison of ozone time series at Ny-Ålesund. On the left are ozone VMR time series of GROMOS-C, OZORAM, ozone sonde, MLS, ACE-FTS, SD-WACCM and ERA5 averaged within 5 pressure intervals where the lower 3 intervals are in the stratosphere and the upper 2 intervals are in the mesosphere. On the right the relative differences to GROMOS-C are shown for the same pressure intervals. The dark blue background indicates polar night and the white background polar day.

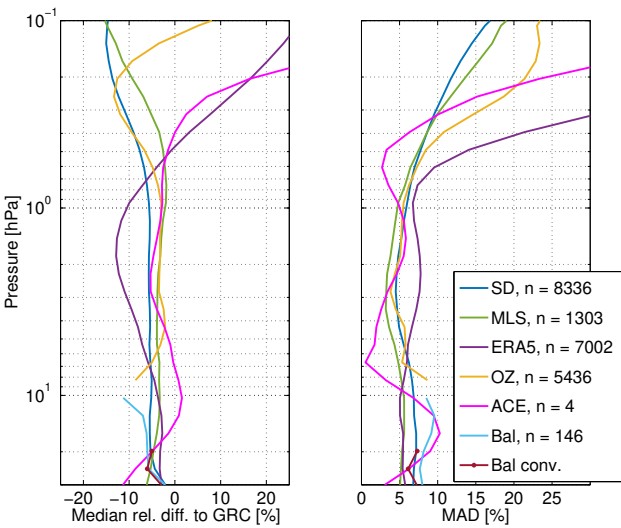

**Figure 6.** Median of the relative difference of SD-WACCM, MLS, ERA5, OZORAM, ACE and balloon-borne ozone sonde profiles and GROMOS-C ozone measurements at Ny-Ålesund (left). Median absolute deviation of the relative difference profiles (right). In the legend $n$ indicates the number of coincident GROMOS-C profiles.

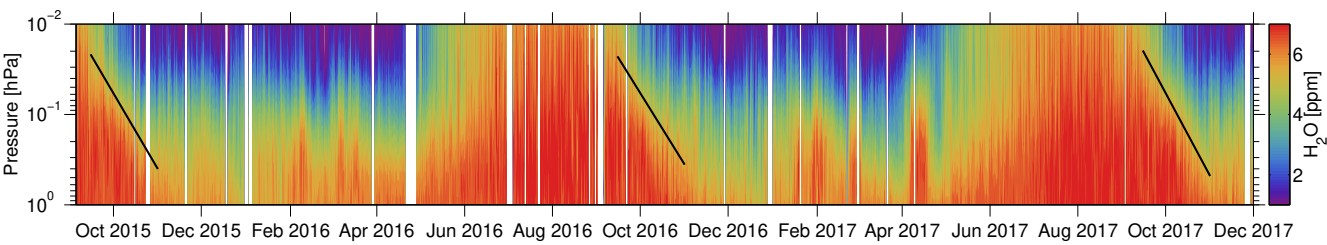

**Figure 7.** MIAWARA-C water vapour time series 2015–2017. The black line indicates the descent rate of water vapour within the polar vortex as derived from a linear fit of the 5.5 ppm isopleth. The descent rates are 428 m/day for 2015, 404 m/day for 2016 and 468 m/day for 2017.

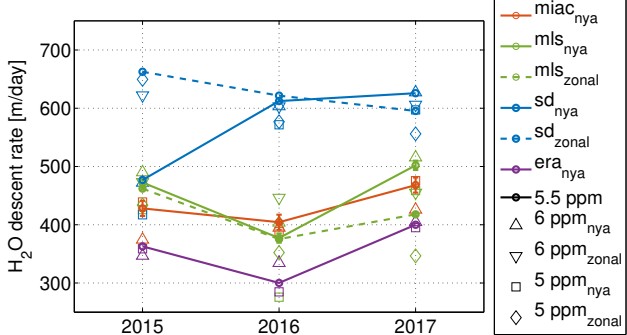

**Figure 8.** Descent rates of water vapour at Ny-Ålesund calculated from the 5 ppm, 5.5 ppm and 6 ppm isopleth from MIAWARA-C, MLS, SD-WACCM and ERA5 between September 15 and November 1 for the years 2015, 2016 and 2017. For MLS and SD-WACCM also the descent rates of the zonally averaged water vapour is shown.

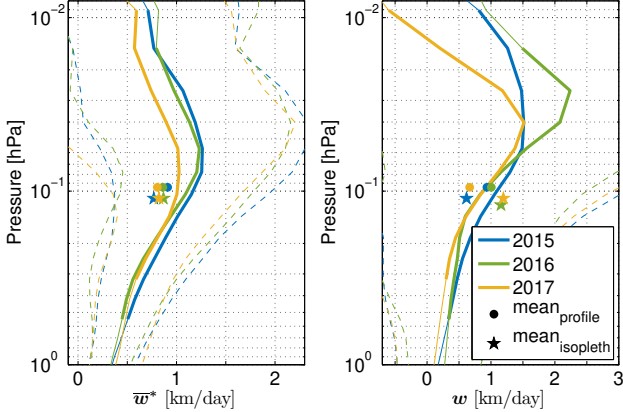

**Figure 9.** Vertical component of the residual mean meridional circulation $\overline{w}^*$ and zonal mean vertical wind $w$ profiles averaged between September 15 and November 1 for the years 2015, 2016 and 2017. The dashed line is the standard deviation and the bold line indicates the altitude range which was covered by the zonal mean 5.5 ppm isopleth of SD-WACCM in the same time period. The points indicate the mean of the profiles over this altitude range and the stars indicate the mean along the zonal mean 5.5 ppm isopleth of SD-WACCM.

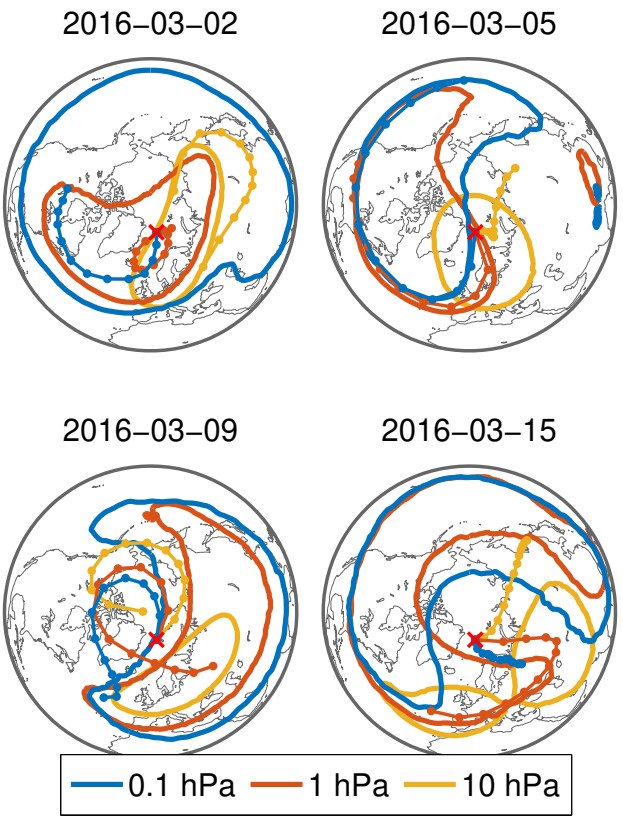

**Figure 10.** The solid lines indicate the contour of the polar vortex at 10 hPa, 1 hPa and 0.1 hPa during the major SSW of 2016. The line with dots shows the LAGRANTO 3-day backward trajectory at the same altitudes. The red cross indicates the location of Ny-Ålesund.

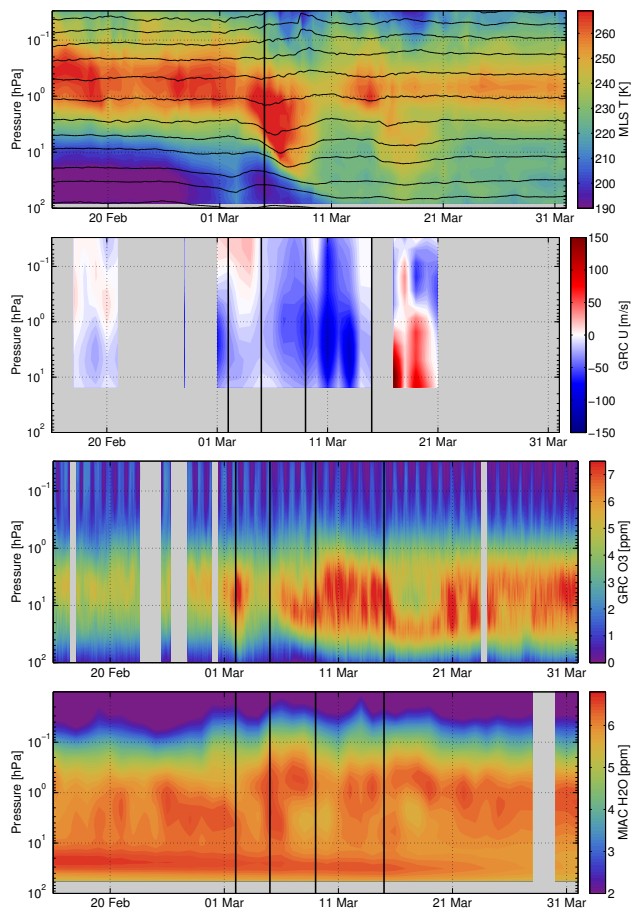

**Figure 11.** Time series of MLS temperature with isolines of potential temperature (solid black lines), GROMOS-C zonal wind, GROMOS-C ozone and MIAWARA-C water vapour during the SSW 2016 at Ny-Ålesund.

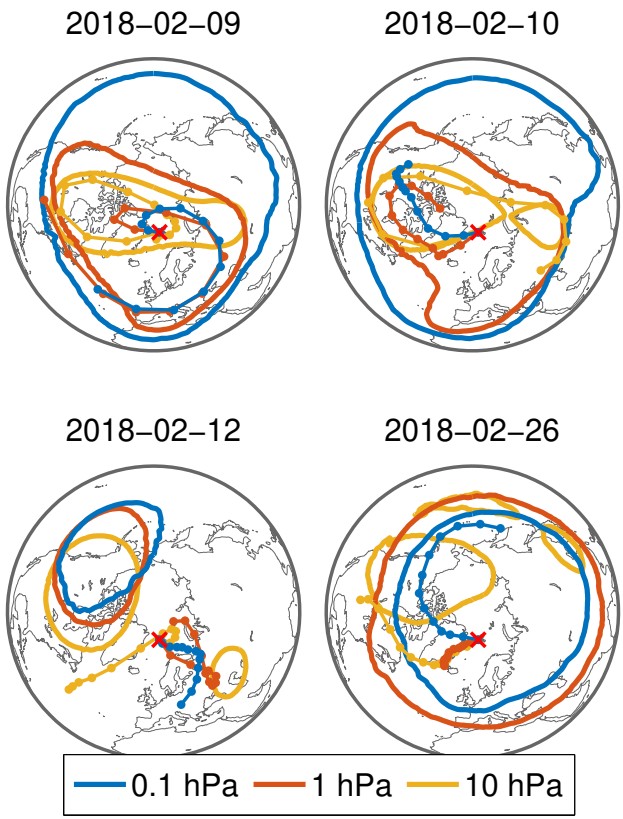

**Figure 12.** The solid lines indicate the contour of the polar vortex at 10 hPa, 1 hPa and 0.1 hPa during the major SSW of 2018. The line with dots shows the LAGRANTO 3-day backward trajectory at the same altitudes. The red cross indicates the location of Ny-Ålesund.

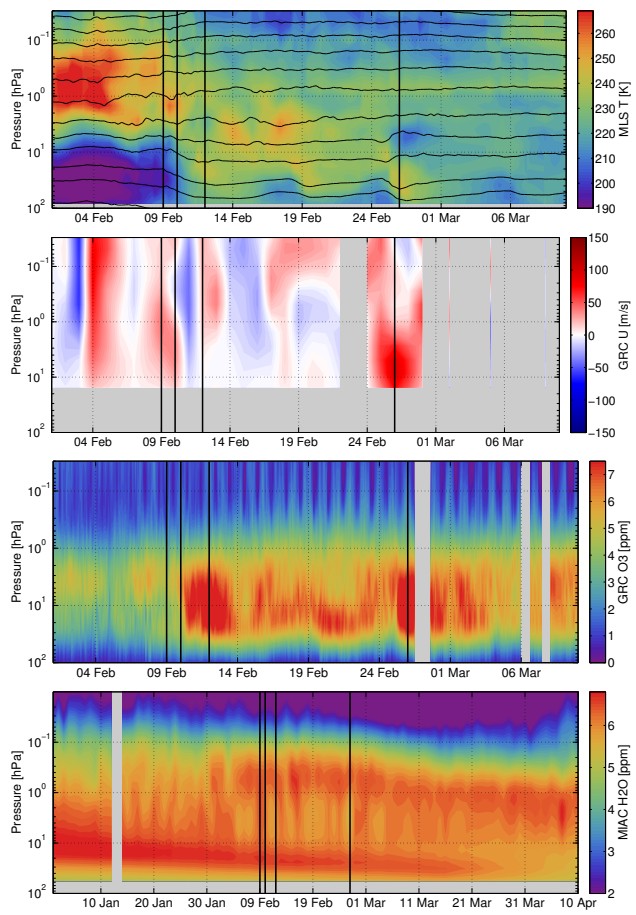

**Figure 13.** Time series of MLS temperature with isolines of potential temperature (solid black lines), GROMOS-C zonal wind, GROMOS-C ozone and MIAWARA-C water vapour during the SSW 2018 at Ny-Ålesund. Note the different time axis for the Water vapour time series.

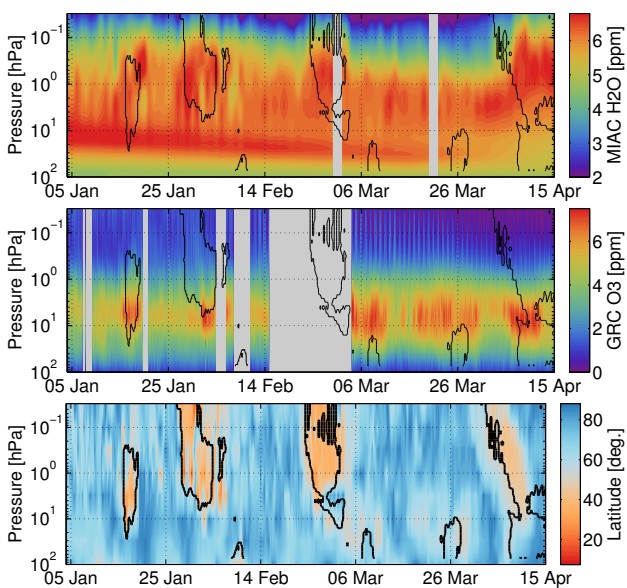

**Figure 14.** Time series of MIAWARA-C water vapour, GROMOS-C ozone and latitudinal origin of the air at Ny-Ålesund calculated with LAGRANTO 3-day backward trajectories. The black contours indicate when the polar vortex edge is above Ny-Ålesund. When the polar vortex shifts away from Ny-Ålesund water vapour increases are measured because airmasses arrive from the midlatitudes. In the ozone measurements an increase in stratospheric ozone during the first two vortex shifts in January and February is visible. In March the ozone increases in the middle stratosphere are less clearly linked to these shifts as during the period of polar night in January/February.

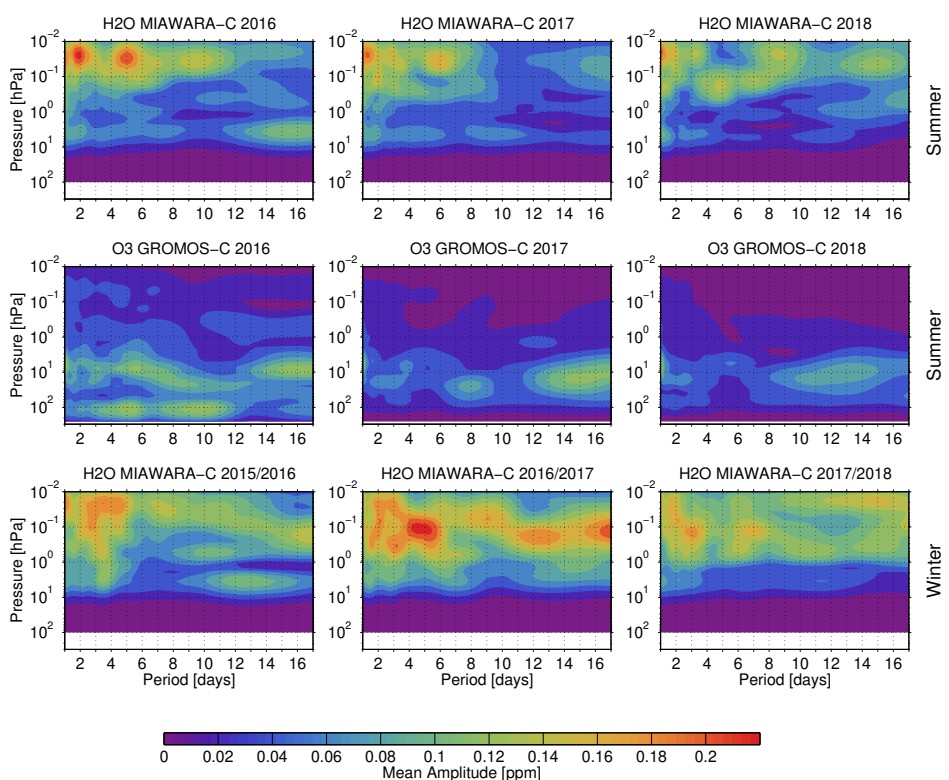

**Figure 15.** Mean amplitude for periods of 1–17 days obtained by bandpass filtering the water vapour and ozone time series for 80 day intervals centred around summer and winter solstice of the years 2015–2018.

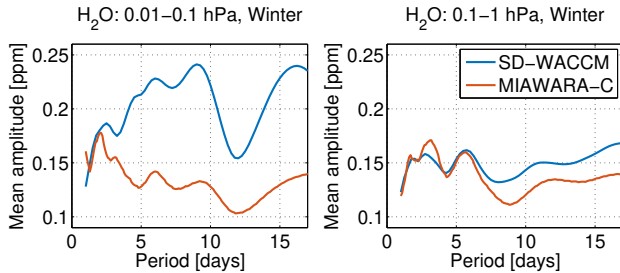

**Figure 16.** Mean amplitude for periods of 1–17 days obtained by bandpass filtering the water vapour time series of MIAWARA-C and SD-WACCM. The signal is averaged for the pressure ranges 0.01–0.1 hPa (left) and 0.1-1 hPa (right) and for $\pm 40$ days around winter solstice of the years 2015–2017.

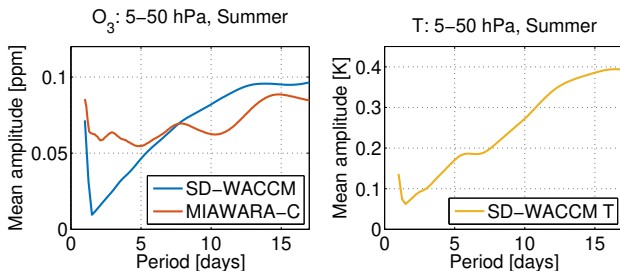

**Figure 17.** Mean amplitude for periods of 1–17 days obtained by bandpass filtering the ozone time series (left) and the temperature time series (right) from GROMOS-C and SD-WACCM. The signal is averaged for 5–50 hPa and for $\pm 40$ days around summer solstice of the years 2016–2018.