# Peer review of "Investigation of Arctic middle-atmospheric dynamics using 3 years of H2O and O3 measurements from microwave radiometers at Ny-Ålesund"

_Atmospheric Chemistry and Physics, 2018_

## Referee Comment (RC1) · Anonymous Referee #1 · 11 Feb 2019

In the present study, Schranz et al. analyse three years of ground-based microwave water vapour and ozone observations in the Arctic stratosphere and compare these data with satellite observations (MLS and ACE-FTS), ERA5 reanalyses and SD-WACCM model simulations. The data are used to study transport and dynamics of the stratosphere and mesosphere, in particular effective mesospheric descent rates, the effects of stratospheric warmings on stratospheric and mesospheric water and ozone and periodic oscillations in the middle atmosphere.

This is a very valuable and important observational data set that can be used to analyse

transport and dynamics of the high latitude middle atmosphere. However, the different topics addressed are not well connected and it is not always clear what has been learned from this analysis. I believe the paper can be strengthened by further exploiting the MLS data as well as the ERA5 reanalyses and the SD-WACCM simulations in comparison with the ground-based observations to arrive at more general conclusions. E.g.:

- In Section 6: Is the effective water vapour descent rate in the mesosphere a good proxy for TEM w_bar_star? In principle the SD-WACCM output should help to address this question.

- Section 7 is very short and I am not at all sure what can be learned from it. Could an analysis of the latitudinal gradients of $H2O$ and $O3$ from MLS (and/or the models) be combined with the ground-based observations to tell us something on the air mass origins that could be contrasted with the Lagranto calculations?

- In Section 9, it didn't become obvious to me what we can learn from the analysed periodicities. Would it make sense to compare these to a corresponding analysis of SD-WACCM to say more directly which of these periodicities are well captured by the model and which are not? Any ideally even why?

Overall I believe that with a few additional analyses of the available data sets and by spelling out more clearly in the text what the conclusions are, the value of these important observations could be further improved. So I recommend publication in Atmos. Chem. Phys. after consideration of the suggested improvements and after addressing my specific comments below.

Specific comments

Abstract, line 1: I have to say, I don't like the expression "dynamic events" too much. This is jargon and not particularly specific. I would suggest to better find a more specific expression, like "dynamics of the polar vortex" or maybe even "factors affecting

subsidence inside the polar vortex".

Abstract, l.8: better spell out (again) what kind of profiles: "the MIAWARA-C profiles" -> "the MIAWARA-C water vapour profiles"

l.10: "Stratospheric GROMOS-C profiles" -> "Stratospheric GROMOS-C ozone profiles"

Abstract: Why is the comparison with SD-WACCM only presented for H2O, not for O3?

Abstract: I suggest to mention NDACC already in the abstract.

p.2, l.18/19: (a) NOx ist produced in the mesosphere not only by solar proton events, but also by energetic electrons. (b) ozone loss of 10% is too specific, numbers could be very different for different events

p.3, l.2: "Atlantic streamer": better spell out (e.g. "steamers of enhanced ozone in the middle stratosphere into the Arctic over the Atlantic sector" if this is what you mean) as the expression does not seem to be standard (yet).

p.4,l.25: "dry bias": better spell out which instrument measures less H2O to avoid any possible misunderstanding

p.5, l.10: Would be nice to have also information on precision and resolution of wind profiles.

Caption Fig. 8: "When the polar vortex shifts away from Ny-Ålesund water vapour and ozone increases are measured because airmasses arrive from the midlatitudes". Ozone increases are not evident from the figure.

p.6, section 3.7: Information on the data sources for water and ozone in ERA-5 is missing. Are these assimilated from (satellite) observations? Or purely modelled?

p.8, l.11: "the diurnal variation is seen in mid summer during the period of polar day": Only during polar day, not during spring and autumn day/night periods?

p.8, l.23: "The balloon borne ozone sonde data were not convolved." Why not? How was this done for the comparison with the OZORAM, which has a similar vertical resolution as the GROMOS?

p.8, l.29: "ERA5 sees": "seeing" does not seem to be an appropriate expression for the assimilated data set.

p.9, l.17 "This annual variation persists up to 1 hPa.": Unclear how this relates to previous sentences.

p.9, l.20: "...deviates substantially from the other datasets and is therefore not included in the intercomparison.": Even if there are substantial differences a comparison would be valuable – in fact how to know that there are differences without a comparison?

p.9, l.28: would be interesting to include also MLS at Ny-Alesund in addition to zonal mean – or investigate the difference between at Ny-Alesund and zonal mean with the models.

p.10, l.25: What does theory tell us about the relation between tracer descent and mean vertical wind? I would have expected that one has to consider a Transformed Eulerian Mean $w\_bar\_star$, instead of just the average vertical wind? Could you calculate $w\_bar\_star$ from SD-WACCM? (see also your own discussion on page 13)

p.10l31/p11,1: "also difficulties with the H2O and CO chemistry": in order to attribute differences to transport or chemistry it would be useful to investigate the relation between tracer descent and, $w\_bar\_star$ and average vertical wind in the model(s)

p.11, l.21: I thought the standard definition for a major warming is a reversal of the 10 hPa zonal mean zonal wind at 60N, not poleward of 60N?

p.12, l.22: "The isentropes show that airmasses were rising in the mesosphere and descending in the stratosphere.": strictly speaking, one should also consider diabatic cooling rates to decide whether air masses are rising together with the isentropes, or descending across the isentropes.

Technical Corrections

p.4, l.30: "a FFT" -> "an FFT"

p.12, l.18: "splitted" -> "split"

---

## Referee Comment (RC2) · Anonymous Referee #2 · 19 Feb 2019

The paper deals with ground-based measurements of water vapor and ozone with a microwave radiometer operated at Ny-Ålesund. The authors retrieve vertical profiles of water vaper and ozone and assess the results by comparing with satellite instruments and model, such as AURA/MLS, ACE-FTS, and SD-WACCM simulation. Almost continuous dataset with a time resolution of hours are obtained since September 2015. Based on the dataset, the authors study dynamical phenomena in the Arctic middle atmosphere including atmospheric descent in the polar vortex, two major SSW events in 2016 and 2018, airmass exchange between the inside and outside of polar vortex

over wide altitude range from stratosphere to mesosphere, and periodicities related to the planetary waves.

The high-time resolution dataset at Ny-Ålesund (79N) in the Arctic region for 3 years is highly valuable, unique, and important particularly for the researchers of polar science. But the analysis method seems basically same as the previous studies by authors' group, and several important details of analysis are missing in the manuscript. It is not clear enough to me what is the originality of this work and what are new findings. The authors could improve the paper by identifying the characteristics at Ny-Ålesund by comparing with the previous studies in Bern and Sodankyla done by authors' group and by presenting them more clearly.

As a whole, the observational results are very interesting and the dataset is invaluable. I think the paper is to be published, but further revisions are necessary before publication to present the key conclusions more clearly. The issues to be considered are raised below.

Specific comments: P8, L9-11: "In the mesosphere a diurnal cycle . . .. . .." The present description about the diurnal variations is too simplified, though the details were already discussed in the author's previous paper, Schranz et al. 2018. At least, the variation pattern (when the VMR is maximum and minimum), typical variation amplitude, and major causes of the variations, are to be described for spring+autumn and polar day, respectively.

P8, L23: "The balloon borne ozone sonde data were not convolved." Why the sonde data are not convolved with the averaging kernel?

P11, L6: "the black contour lines indicate when the polar vortex edge is right above Ny-Ålesund . . .. . .. .." There is no explanation how the black lines are drawn. How the authors determine the polar vortex edge? What are the criteria?

P11, L10: "Lagranto" appears for the first time. So, brief explanation about Lagranto is

necessary.

P11, L27-28: "The polar vortex edge was derived from . . . . . . (Scheiben et al. 2012)." I think the polar vortex edge is conventionally defined in terms of potential vorticity and/or equivalent latitude. There must be small discussions about the appropriateness of the definition in this paper, i.e. the polar vortex edge determined by the GPH contours and the maximum wind velocity, though detailed discussion would be described in Scheiben et al. 2012.

P12, L10: eastward ==> westward ??

P12, L28-29: "Ongoing meridional mixing brought again ozone rich air from the midlatitudes on February 25 and the ozone VMR increased again to 8 ppm." Is this statement concluded by trajectory analysis or speculation? There are no supporting materials for this argument. It is difficult to conclude this only from the series of vortex edge snapshots in Figure 11. On the other hand, the top panel of Figure 12 shows that the temperature is suddenly decreasing around 10 hPa on Feb 25. According to the ozone chemistry, ozone VMR tends to increase if the temperature decreases. Does this temperature-change contribute to the ozone increase or the effect is negligible? Please make more quantitative and more careful discussion by using trajectory analysis and ozone distribution in the mid-latitude obtained by MLS and/or SD-WACCM.

P13, L12: "because midlatitude air is drier than vortex air at that altitude." Why the mid latitude air is drier? Please explain this more detail by using appropriate references.

P13, L27-28: "5.2 water vapor isopleth" → "5.2 ppm water vapor isopleth".

Figure 13: On the vertical axis, there is only one tick at 10ˆ0 hPa. At least two ticks and labels are necessary to indicate the span.

---

## Author Comment (AC1) · 7 May 2019

**Response to Referees #1 and #2**

Franziska Schranz, Brigitte Tschanz, Rolf Rüfenacht, Klemens Hocke,
Mathias Palm, Niklaus Kämpfer

May 7, 2019

The authors thank Referee #1 and Referee #2 for their encouraging comments and especially for the suggestions on how to improve the general conclusions.

In the following the specific comments of Referee #1 and Referee #2 are addressed and an answer to the general comments is given. The technical corrections have been made.

**Response to the specific comments of Referee #1**

1)
Abstract, line 1: I have to say, I dont like the expression dynamic events too much. This is jargon and not particularly specific. I would suggest to better find a more specific expression, like dynamics of the polar vortex or maybe even factors affecting subsidence inside the polar vortex.

The authors agree and the sentence was changed.

We used 3 years of water vapour and ozone measurements to study the dynamics in the Arctic middle atmosphere. We investigated the descent of water vapour within the polar vortex, major and minor sudden stratospheric warmings and periodicities at Ny-Ålesund.

2)
Abstract, l.8: better spell out (again) what kind of profiles: the MIAWARA-C profiles → the MIAWARA-C water vapour profiles
Abstract, l.10: Stratospheric GROMOS-C profiles → Stratospheric GROMOS-C ozone profiles

The suggested changes have been made.

3)
Abstract: Why is the comparison with SD-WACCM only presented for H2O, not for O3?

We see the importance of showing the SD-WACCM comparison for both H2O and O3 and added the result for O3.

Stratospheric GROMOS-C ozone profiles are on average within 6 % of the model SD-WACCM, the satellite instruments AuraMLS and ACE-FTS and the ground-based microwave radiometer OZORAM which is also located at Ny-Ålesund.

4)
Abstract: I suggest to mention NDACC already in the abstract.

The following sentence has been added:

Both instruments belong to the Network for the Detection of Atmospheric Composition Change (NDACC).

5)
p.2, l.18/19: (a) NOx ist produced in the mesosphere not only by solar proton events, but also by energetic electrons. (b) ozone loss of 10% is too specific, numbers could be very different for different events

We agree that this statement was to specific and we made a more general statement.

This includes long lived constituents like NOx and HOx which lead to ozone destruction in the mesosphere and stratosphere (eg. Randall et al., 2009). Through this mechanism energetic particle precipitation, which produces NOx and HOx in the mesosphere and lower thermosphere, has an influence on polar ozone (Andersson et al., 2018).

6)
p.3, l.2: Atlantic streamer: better spell out (e.g. steamers of enhanced ozone in the middle stratosphere into the Arctic over the Atlantic sector if this is what you mean) as the expression does not seem to be standard (yet).

Yes this is what we mean. We changed the sentence accordingly.

Together with water vapour it was used as a tracer for vortex filamentation in the lower stratosphere (Müller et al., 2003) and for streamers of enhanced ozone in the middle stratosphere along the edge of the polar vortex over the Atlantic sector (Hocke et al., 2017).

7)
p.4,l.25: dry bias: better spell out which instrument measures less H2O to avoid any possible misunderstanding

Thank you for pointing this out. The sentence has been rewritten:

With respect to MLS version 3.3, there is almost no bias at the lowest altitude level (4 hPa) but with higher altitudes the bias increases and at 0.02 hPa MIAWARA-C measures up to 13 % less water vapour than MLS.

8)
p.5, l.10: Would be nice to have also information on precision and resolution of wind profiles.

We agree and added the following two sentences.

The wind profiles range from about 40 up to 60–70 km with a vertical resolution of 10–20 km.

A comparison with the ground-based wind radiometer WIRA (Rüfenacht et al., 2014) at La Reunion shows that GROMOS-C captures the principal wind features (Fernandez et al., 2016).

9)
Caption Fig. 8: When the polar vortex shifts away from Ny-Ålesund water vapour and ozone increases are measured because airmasses arrive from the midlatitudes. Ozone increases are not evident from the figure.

We agree that the increases in ozone are not as evident as in water vapour. We changed the caption to be more precise.

When the polar vortex shifts away from Ny-Ålesund water vapour increases are measured because airmasses arrive from the midlatitudes. In the ozone measurements an increase in stratospheric ozone during the first two vortex shifts in January and February is visible. In March the ozone increases in the middle stratosphere are less clearly linked to these shifts as during the period of polar night in January/February.

10)
p.6, section 3.7: Information on the data sources for water and ozone in ERA-5 is missing. Are these assimilated from (satellite) observations? Or purely modelled?

Both, ozone and water vapour, are assimilated from observations. We added the following sentences:

For the considered time period ozone is assimilated from retrievals of the GOME-2 instruments on the METOP-A/B satellites, the SBUV-2 instruments on the NOAA satellites and of MLS and OMI on the EOS-Aura satellite. Water vapour is assimilated from humidity profiles measured with radiosondes and from ground stations which are provided by the World Meteorological Organizations Information System (WMO WIS).

11)
p.8, l.11: the diurnal variation is seen in mid summer during the period of polar day: Only during polar day, not during spring and autumn day/night periods?

This depends on the altitude. At 10 hPa pronounced diurnal ozone variations are seen from May until the end of August. Referee #2 asked for a more detailed description of the diurnal cycle and therefore we changed the manuscript as follows:

The measurements of the year 2016 have been used to study the diurnal ozone variations throughout the year (Schranz et al., 2018). In the mesosphere a diurnal cycle was detected in spring and autumn when there is light and darkness within one day. Ozone is depleted through photodissociation during daytime and subsequently recombines at night which leads to a diurnal ozone variation of up to 1 ppm at 0.1 hPa. In the stratosphere the diurnal variations are seen throughout the polar day. At 10 hPa the largest variations of about 0.3 ppm are seen around summer solstice. At this altitude the net ozone production is positive for a solar zenith angle smaller than 65–75°, depending on the season, otherwise the net production is negative which leads to an ozone maximum in the late afternoon. At 1 hPa around the stratopause the diurnal cycle has the largest amplitudes of 0.5 ppm in the end of April/beginning of May and in August.

The chemistry of diurnal ozone variations in general is described in Schanz et al. (2014) and specifically for Ny-Ålesund in Schranz et al. (2018).

12)
p.8, l.23: The balloon borne ozone sonde data were not convolved. Why not? How was this done for the comparison with the OZORAM, which has a similar vertical resolution as the GROMOS?

Convolving a profile means that for a given altitude measurements from higher and lower altitudes contribute to the convolved data point according to the averaging kernel. The balloon sondes reach only up to an altitude of approximately 10 hPa which means that the topmost data points can not be convolved correctly. The threshold for a "good" convolution was set such that at least 80% of the absolute area of the corresponding averaging kernel needs to be below 10 hPa. This leads to only 2 data points being within the region of interest. We decided to show the difference profiles with both the convolved and the unconvolved measurement. Originally we convolved the OZORAM profiles but upon your question we decided to use the unconvolved profiles. This does almost not affect the stratospheric part but in the mesosphere the relative difference increases by up to 10%. Figure 6 was adapted accordingly.

The OZORAM data were not convolved because the vertical resolution is comparable to the one of GROMOS-C. For the balloon borne ozone sonde measurements the relative difference for both the convolved and unconvolved data are shown because a meaningful convolution is only possible up to 20 hPa.

13)
p.8, l.29: ERA5 sees: "seeing does not seem to be an appropriate expression for the assimilated data set.

We agree and have changed the word.

ERA5 models less water vapour in late summer but is also mostly within ±10 % of MIAWARA-C.

14)
p.9, l.17 This annual variation persists up to 1 hPa.: Unclear how this relates to previous sentences.

We mean the annual variation of the relative differences to GROMOS-C. In summer all the other datasets detect less ozone than GROMOS-C whereas in winter they agree mainly within 10 %.

In the lowest panel (30–10 hPa) all models and instruments agree with each other except for GROMOS-C which measures up to 20 % higher ozone VMRs in summer whereas in winter it is mainly within 10 % of the other datasets. This annual variation of the relative differences to GROMOS-C persists up to 1 hPa.

15)
p.9, l.20: . . .deviates substantially from the other datasets and is therefore not included in the intercomparison.: Even if there are substantial differences a comparison would be valuable  in fact how to know that there are differences without a comparison?

In the model which is used for the ERA5 reanalysis ozone is parametrized. The parametrization focuses on the stratosphere and ozone in the mesosphere is therefore not well captured. We agree that this intercomparison is nonetheless of interest and we updated Figure 5 and 6 and discussed the deviation.

Up to 3 hPa ERA5 ozone VMR agrees well with the other datasets but above it starts to deviate, mainly during summer but also in winter at 0.3–0.1 hPa. This is because the model which is used for the ERA5 reanalysis has no interactive ozone chemistry but uses a parametrization (Cariolle and Teyss, 2007) and there is no ozone assimilation in the mesosphere.
Up to 0.5 hPa (about 55 km) the median of the differences relative to GROMOS-C is mainly within 5 % for OZORAM (above 6 hPa), MLS, ACE, SD-WACCM and ERA5 (Fig. 6). The relative difference of the balloon borne ozone sonde to MIAWARA-C is increasing from -3 % at 30 hPa to -13 % at 10 hPa. In general the ozone measurements of GROMOS-C are up to 5 % higher than OZORAM, MLS, SD-WACCM and ERA5.

16)
p.9, l.28: would be interesting to include also MLS at Ny-Ålesund in addition to zonal mean or investigate the difference between at Ny-Ålesund and zonal mean with the models.

We added an additional Figure where we show the water vapour descent rates of MIAWARA-C, MLS, SD-WACCM and ERA5 for Ny-Ålesund. For MLS and SD-WACCM we also show the zonal mean descent rate. The descent rates are shown for the 5.5 ppm water vapour isopleth and additionally for the 5 and 6 ppm isopleth.

In Fig. 8 the water vapour descent rates from MIAWARA-C are compared to the descent rates of MLS, SD-WACCM and ERA5. The solid line connects descent rates from the 5.5 ppm isopleth at Ny-Ålesund and the dashed line indicates zonal mean descent rates. Descent rates from the 5 and 6 ppm isopleths are indicated with different symbols. The descent rates from the MLS water vapour measurements at Ny-Ålesund and for the zonal mean are within 10 % of MIAWARA-C. The models do however have an average discrepancy of +30 % for SD-WACCM and -20 % for ERA5 at Ny-Ålesund. The average discrepancy for the zonal mean of SD-WACCM is 45 %.

17)
p.10, l.25: What does theory tell us about the relation between tracer descent and mean vertical wind? I would have expected that one has to consider a Transformed Eulerian Mean w_bar_star, instead of just the average vertical wind? Could you calculate w_bar_star from SD-WACCM? (see also your own discussion on page 13)

Thank you for this comment. We calculated $\overline{w}^*$ from the SD-WACCM data and found a good agreement between the zonal mean water vapour descent rate of SD-WACCM and $\overline{w}^*$ if it was averaged along the isopleth which was used to calculate the water vapour descent rate.

The SD-WACCM simulations show that $\overline{w}^*$, if it is averaged along the 5.5 ppm isopleth of MIAWARA-C, is within -4–26 % of the zonal mean water vapour descent rates. This shows that in general water vapour is a reasonable proxy for the vertical bulk motion in the high Arctic during the formation of the polar vortex in autumn. The difference of -4–26 % shows however that other processes than vertical advection contribute to the effective descent rate of water vapour in the polar vortex.

18)

p.10l31/p11,1: also difficulties with the H2O and CO chemistry: in order to attribute differences to transport or chemistry it would be useful to investigate the relation between tracer descent and, w_bar_star and average vertical wind in the model(s)

We also added a Figure with velocity profiles of $\overline{w}^*$ and the vertical wind. And we calculated the mean velocities along the isopleths to intercompare the volocities with the water vapour descent rates.

Figure 9 shows mean profiles of the vertical component of the mean meridional circulation $\overline{w}^*$ and the vertical wind from SD-WACCM for September 15. until November 1. of the years 2015, 2016 and 2017. The maximum of $\overline{w}^*$ is at around 7 hPa with 1000–1350 m/day, towards 1 hPa it decreases to 350–400 m/day. The bold line in Fig. 9 indicates the altitude range covered by the 5.5 ppm water vapour isopleth of MIAWARA-C and the points indicate the mean over these altitude ranges. The averaged $\overline{w}^*$ range from 800 to 1080 m/day which is substantially higher than the water vapour descent rate from SD-WACCM. However if we average $\overline{w}^*$ along the 5.5 ppm isopleths of MIAWARA-C the velocities differ only by 12 %, 26 % and -4 % from the zonal mean water vapour descent rates of SD-WACCM. The averaged mean vertical wind profiles are very close to $\overline{w}^*$ whereas the profiles show a higher descent rate in the upper mesosphere and a smaller descent rate in the lower mesosphere.

19)

p.11, l.21: I thought the standard definition for a major warming is a reversal of the 10 hPa zonal mean zonal wind at 60N, not poleward of 60N?

There exist several definitions of sudden stratospheric warmings (SSW) (see eg. Butler et al., 2015). A popular definition for major SSWs is given in Charlton and Polvani (2007). They use only the reversal of the zonal mean zonal wind at 60 °N and 10 hPa as a criterion. We decided to use the WMO definition of McInturff (1978) because also minor SSWs are defined. For a major SSW the zonal mean temperature gradient from 60–90°N needs to be positive at 10 hPa or below and the zonal mean zonal wind needs to reverse poleward of 60°N.

20)

p.12, l.22: The isentropes show that airmasses were rising in the mesosphere and descending in the stratosphere.: strictly speaking, one should also consider diabatic cooling rates to decide whether air masses are rising together with the isentropes, or descending across the isentropes.

This is true. Instead of the cooling rates we looked at the $\overline{w}^*$ where we found strong upward motion in the mesosphere from about February 10.–19. and downward motion in the stratosphere from about February 10.–15., 2018. We changed the statement as follows:

The rise of the isentropes in the mesosphere and the descent in the stratosphere indicate a corresponding motion of airmasses which is actually found in the vertical velocity $\overline{w}^*$ of the mean residual circulation.

**Response to the specific comments of Referee #2**

1)

P8, L9-11: In the mesosphere a diurnal cycle ... The present description about the diurnal variations is too simplified, though the details were already discussed in the authors previous paper, Schranz et al. 2018. At least, the variation pattern (when the VMR is maximum and minimum), typical variation amplitude, and major causes of the variations, are to be described for spring+autumn and polar day, respectively.

We added the additional information about the diurnal cycle:

The measurements of the year 2016 have been used to study the diurnal ozone variations throughout the year (Schranz et al., 2018). In the mesosphere a diurnal cycle was detected in spring and autumn when there is light and darkness within one day. Ozone is depleted through photodissociation during daytime and subsequently recombines at night which leads to a diurnal ozone variation of up to 1 ppm at 0.1 hPa. In the stratosphere the diurnal variations are seen throughout the polar day. At 10 hPa the largest variations of about 0.3 ppm are seen around summer solstice. At this altitude the net ozone production is positive for a solar zenith angle smaller than 65–75°, depending on the season, otherwise the net production is negative which leads to an ozone maximum in the late afternoon. At 1 hPa around the stratopause the diurnal cycle has the largest amplitudes of 0.5 ppm in the end of April/beginning of May and in August. The chemistry of diurnal ozone variations in general is described in Schanz et al. (2014) and specifically for Ny-Ålesund in Schranz et al. (2018).

2)

P8, L23: The balloon borne ozone sonde data were not convolved. Why the sonde data are not convolved with the averaging kernel?

Convolving a profile means that for a given altitude measurements from higher and lower altitudes contribute to the convolved data point according to the averaging kernel. The balloon sondes reach only up to an altitude of approximately 10 hPa which means that the topmost data points can not be convolved correctly. The threshold for a "good" convolution was set such that at least 80% of the absolute area of the corresponding averaging kernel needs to be below 10 hPa. This leads to only 2 data points being within the region of interest. We decided to show the difference profiles with both the convolved and the unconvolved measurement.

For the balloon borne ozone sonde measurements the relative difference for both the convolved and unconvolved data are shown because a meaningful convolution is only possible up to 20 hPa.

3)

P11, L6: the black contour lines indicate when the polar vortex edge is right above Ny-Ålesund ... There is no explanation how the black lines are drawn. How the authors determine the polar vortex edge? What are the criteria?

We calculate the edge of the polar vortex as described under point 5). We then determined for every pressure level and every 6-hour time step if Ny-Ålesund is in or outside of the polar vortex. The black line was then drawn as the contour line of this inside/outside field.

For every model level and every 6-hour time step we calculated if Ny-Ålesund is in or outside

of the polar vortex. The contour line of this inside/outside field indicates then the polar vortex edge was passing Ny-Ålesund.

4)
P11, L10: Lagranto appears for the first time. So, brief explanation about Lagranto is necessary.

We added a brief description and a reference.

Backward trajectories are calculated with the lagrangian analysis tool (LAGRANTO, Sprenger and Wernli, 2015) using wind fields from the ECMWF operational data.

5)
P11, L27-28: The polar vortex edge was derived from . . . (Scheiben et al. 2012). I think the polar vortex edge is conventionally defined in terms of potential vorticity and/or equivalent latitude. There must be small discussions about the appropriateness of the definition in this paper, i.e. the polar vortex edge determined by the GPH contours and the maximum wind velocity, though detailed discussion would be described in Scheiben et al. 2012.

We agree and added a discussion about the appropriateness of the polar vortex definition.

For the discussion of the SSWs we determined the edge of the polar vortex from ECMWF geopotential height (GPH) and wind fields as the GPH contours with highest wind speed at a given pressure level. This definition for the polar vortex edge is used because it performs well from the stratosphere up to the mesosphere and even during an SSW. This is shown in Scheiben et al. (2012) where the method is also discussed in detail. Another possibility to define the polar vortex edge is to use the maximum of the potential vorticity gradients along potential vorticity isolines (Nash et al., 1996). Potential vorticity is an excellent tracer for the polar vortex edge in the stratosphere. It is however no longer a vortex centred coordinate above 60 km (Harvey et al., 2009) and can not be used to determine the polar vortex edge in the mesosphere.

6)
P12, L10: eastward → westward ??

Yes, this was changed.

7)
P12, L28-29: Ongoing meridional mixing brought again ozone rich air from the midlatitudes on February 25 and the ozone VMR increased again to 8 ppm. Is this statement concluded by trajectory analysis or speculation? There are no supporting materials for this argument. It is difficult to conclude this only from the series of vortex edge snapshots in Figure 11. On the other hand, the top panel of Figure 12 shows that the temperature is suddenly decreasing around 10 hPa on Feb 25. According to the ozone chemistry, ozone VMR tends to increase if the temperature decreases. Does this temperature-change contribute to the ozone increase or the effect is negligible? Please make more quantitative and more careful discussion by using trajectory analysis and ozone distribution in the mid-latitude obtained by MLS and/or SD-WACCM.

This statement was concluded by trajectory analysis. We added the trajectories to the series of the polar vortex contours in Fig. 9 and 11 to support this statement. At 10 hPa the net chemical

production rate of ozone is negative during the polar night and is first positive in the beginning of April. During the temperature decrease around Feb 25. the net chemical ozone production rate was still negative which means that the increase in ozone is solely due to transport. In the Arctic, in contrast to the midlatitudes, there is no anticorrelation detected between temperature and ozone in winter.

Ongoing meridional mixing brought again ozone rich air from the centre of the United States on February 25 and the ozone VMR increased again to 8 ppm.

At Ny-Ålesund the net chemical ozone production rate in the stratosphere, taken from SD-WACCM, is always negative during the periods when the 2 SSWs took place and the enhancements result therefore solely from transport of ozone rich midlatitude air to the pole. The ozone increase of February 25 is therefore not a result of the decrease in temperature which was observed at the same time.

8)
P13, L12: because midlatitude air is drier than vortex air at that altitude. Why the mid latitude air is drier? Please explain this more detail by using appropriate references.

The mechanism has already been introduced in the introduction. We additionally added a short explanation and a reference.

The midlatitude air also brought moister air to the mesosphere at Ny-Ålesund whereas in the stratosphere (around 3 hPa) water vapour decreased. Midlatitude air is drier than vortex air at that altitude because of the subsidence of water vapour rich airmasses from higher stratospheric levels inside of the polar vortex (Lossow et al., 2009).

9)
P13, L27-28: 5.2 water vapor isopleth  5.2 ppm water vapor isopleth.

We agree and the unit was inserted.

10)
Figure 13: On the vertical axis, there is only one tick at 100 hPa. At least two ticks and labels are necessary to indicate the span.

Thank you for highlighting this. We adapted the figure accordingly.

**Response to the general comments of Referee #1 and Referee #2**

Chapter 7: Effective descent rate of $H_2O$
We calculated the mean residual circulation from SD-WACCM and found that the water vapour descent rate agrees within -4–26 % with $\overline{w}^*$ averaged along the isopleth which was used for the calculation of the descent rate. We conclude that water vapour is a good proxy for the descent rate inside of the polar vortex in autumn although vertical advection is not the only factor contributing to the effective descent rate of water vapour. See also the answers to the comments 16–19.

Chapter 8: Vortex shifts in winter 2017

We combined the chapters 8 and 9 and the chapter about the vortex shifts is now part of the analysis of major and minor warmings. With the analysis of the minor warmings in 2017 we want to highlight that the separation of midlatitude and polar air by the polar vortex persists during minor warmings which is in contrast to the total breakdown of the stratospheric polar vortex which was seen during the two major SSWs and that both water vapour and ozone act as a tracer for polar vortex air which is confirmed by the trajectory analysis.

Chapter 9: Periodicities

In Chapter 9 we give an overview of the periodicities seen in the water vapour an ozone time series from Ny-Ålesund and we show the seasonal and interannual differences in the amplitude spectra. We find that in summer the atmospheric waves have largest amplitudes at higher altitudes than in winter. We additionally added a comparison with SD-WACCM. For water vapour in winter we find that SD-WACCM captures the wave activity nicely between 0.1 and 1 hPa. However at higher altitudes (0.1–0.01 hPa) SD-WACCM and MIAWARA-C show completely different spectral amplitude patterns. For ozone we looked at the summer stratosphere. SD-WACCM does not capture the peaks around 8 and 15 days but the general structure with increasing amplitude with growing period length is seen. The temperature shows a similar amplitude spectrum as SD-WACCM ozone and the periodicities in ozone might be driven by periodicities in temperature.

Comparison with previous work at Bern and Sodankylä

In the conclusion we highlighted the different behaviour of ozone in the midlatitudes and the Arctic. Ozone in the Arctic is increasing during SSWs because of ozone rich air which is transported to the Pole from the midlatitudes. Whereas in the midlatitudes the increasing temperatures contribute to ozone depletion. Further we mention in the conclusions that the bias of MLS to MIAWARA-C was already seen in 2011-2013 when MIAWARA-C was located at Bern and Sodankylä.

**References**

Andersson, M. E., Verronen, P. T., Marsh, D. R., Seppälä, A., Päivärinta, S.-M., Rodger, C. J., Clilverd, M. A., Kalakoski, N., and van de Kamp, M.: Polar Ozone Response to Energetic Particle Precipitation Over Decadal Time Scales : The Role of Medium-Energy Electrons, J. Geophys. Res. Atmos., 123, 607–6022, doi:10.1002/2017JD027605, 2018.

Butler, A. H., Seidel, D. J., Hardiman, S. C., Butchart, N., Birner, T., and Match, A.: Defining sudden stratospheric warmings, Bull. Am. Meteorol. Soc., 96, 1913–1928, doi:10.1175/BAMS-D-13-00173.1, 2015.

Cariolle, D. and Teyss, H.: A revised linear ozone photochemistry parameterization for use in transport and general circulation models : multi-annual simulations, Atmos. Chem. Phys., 7, 2183–2196, 2007.

Charlton, A. J. and Polvani, L. M.: A New Look at Stratospheric Sudden Warmings . Part I : Climatology and Modeling Benchmarks, Am. Meteorol. Soc., 20, 449–469, 2007.

Fernandez, S., Rüfenacht, R., Kämpfer, N., Portafaix, T., and Posny, F.: Results from the validation campaign of the ozone radiometer GROMOS-C at the NDACC station of Réunion island, Atmos. Chem. Phys., 16, 7531–7543, doi:10.5194/acp-16-7531-2016, 2016.

Harvey, V. L., Randall, C. E., and Hitchman, M. H.: Breakdown of potential vorticity based equivalent latitude as a vortex-centered coordinate in the polar winter mesosphere, J. Geophys. Res., 114, 1–12, doi:10.1029/2009JD012681, 2009.

Hocke, K., Schranz, F., Barras, E. M., Moreira, L., and Kämpfer, N.: An Atlantic streamer in stratospheric ozone observations and SD-WACCM simulation data, Atmos. Chem. Phys., 17, 3445–3452, doi:10.5194/acp-17-3445-2017, 2017.

Lossow, S., Khaplanov, M., Gumbel, J., Stegman, J., Witt, G., Dalin, P., Kirkwood, S., Schmidlin, F. J., Fricke, K. H., and Blum, U.: Middle atmospheric water vapour and dynamics in the vicinity of the polar vortex during the Hygrosonde-2 campaign, Atmos. Chem. Phys., 9, 4407–4417, 2009.

McInturff, R. M.: Stratospheric warmings: Synoptic, dynamic and general-circulation aspects, Natl. Aeronaut. Sp. Adm. Sci. Tech. Inf. Off., URL http://hdl.handle.net/2060/19780010687, 1978.

Müller, M., Neuber, R., Fierli, F., Hauchecorne, A., Vömel, H., and Oltmans, S. J.: Stratospheric water vapour as tracer for vortex filamentation in the arctic winter 2002/2003, Atmos. Chem. Phys., 3, 1991–1997, doi:10.5194/acp-3-1991-2003, 2003.

Nash, E. R., Newman, P. A., Rosenfield, J. E., and Schoeberl, M. R.: An objective determination of the polar vortex using Ertel' s potential vorticity, J. Geophys. Res., 101, 9471–9478, 1996.

Randall, C. E., Harvey, V. L., Siskind, D. E., France, J., Bernath, P. F., Boone, C. D., and Walker, K. A.: NOx descent in the Arctic middle atmosphere in early 2009, Geophys. Res. Lett., 36, 1–4, doi:10.1029/2009GL039706, 2009.

Rüfenacht, R., Murk, A., Kämpfer, N., Eriksson, P., and Buehler, S. A.: Middle-atmospheric zonal and meridional wind profiles from polar, tropical and midlatitudes with the ground-based microwave Doppler wind radiometer WIRA, Atmos. Meas. Tech., pp. 4491–4505, doi: 10.5194/amt-7-4491-2014, 2014.

Schanz, A., Hocke, K., and Kämpfer, N.: Daily ozone cycle in the stratosphere : global , regional and seasonal behaviour modelled with the Whole Atmosphere Community Climate Model, Atmos. Chem. Phys., 14, 7645–7663, doi:10.5194/acp-14-7645-2014, 2014.

Scheiben, D., Straub, C., Hocke, K., Forkman, P., and Kämpfer, N.: Observations of middle atmospheric H2O and O3 during the 2010 major sudden stratospheric warming by a network of microwave radiometers, Atmos. Chem. Phys., 12, 7753–7765, doi:10.5194/acp-12-7753-2012, 2012.

Schranz, F., Fernandez, S., Kämpfer, N., and Palm, M.: Diurnal variation in middle atmospheric ozone by ground based microwave radiometry at Ny-Ålesund over 1 year, Atmos. Chem. Phys., 18, 4113–4130, doi:10.5194/acp-2017-1080, 2018.

Sprenger, M. and Wernli, H.: The LAGRANTO Lagrangian analysis tool version 2.0, Geosci. Model Dev., 8, 2569–2586, doi:10.5194/gmd-8-2569-2015, 2015.

---

## Referee Report (RR1)

$2^{nd}$ report on the revised manuscript "Investigation of Arctic middle-atmospheric dynamics using 3 years of $H_2O$ and $O_3$ measurements from microwave radiometers at Ny-Ålesund" by Schranz et al.

The revised manuscript was reorganized and properly improved by taking most of reviewers' comments into account. Complementary explanations and new supplemental analyses were added appropriately. Particularly, comparison between the apparent descending velocity of water vapor and the vertical velocity of the residual mean meridional circulation, w_star_bar, suggested by the referee #1 is a meaningful argument. There still remains large discrepancies between the both velocities, but the authors made detailed analyses and described the breakdown of the discrepancy; that the both velocities are agreed within 16-39% after some averaging processes within the SD-WACCM model, and that there is an average discrepancy of ~ 30 % between the observed velocity derived from MIAWARA-C and that from the SD-WACCM model. I think that complete understanding of the causes of the discrepancies is difficult only by the authors and that further discussions by the research community including specialists of model development and/or theoretical researchers of atmospheric dynamics are necessary.

The observed data and the analysis methods were properly presented, and the previous studies are well reviewed and compared with in the revised manuscript. The results and arguments are valuable enough to stimulate and activate the future discussion in the research community. Thus I recommend publication of this manuscript after minor revisions.

Minor and technical points:

Page 11, Line 31-32

"The relative difference …. Ozone sonde to MIAWARA-C……" ➔

I think "GROMOS-C" is correct instead of "MIAWARA-C".

Page 12, Line 18

"at around 7 hPa" ➔ "at around 0.07 hPa"

Page 12, Line 23

"averaged mean vertical wind profiles" ➔ "averaged zonal mean vertical wind profiles"

Page 30, the last sentence of Figure 9 caption

According to the main text in Page 18, line 18, the w_star_bar is averaged along the fit of the 5.5 ppm isopleths of SD-WACCM.   So, the appropriate expression of the last sentence is "The points indicate the mean of the profiles over the altitude range of isopleth of MIAWARA-C, and the stars indicate the mean along the fit of the 5.5ppm isopleths of SD-WACCM", isn't it?

---

## Author Response (AR3)

**Response to the second review of Referees #1 and #2**

**Franziska Schranz, Brigitte Tschanz, Rolf Rüfenacht, Klemens Hocke, Mathias Palm, Niklaus Kämpfer**

**July 5, 2019**

The authors thank Referee #1 and Referee #2 for their second review and for the careful reading of the manuscript. The technical corrections have been made and are listed below. Subsequently we address 3 comments in more detail. Page and line numbers refer to the document with tracked changes.

**Technical corrections**

Page 2, Line 3: are is → are

Page 2, Line 5: You need to carefully consider present and past form here. I believe We further presented should be We further present, but what about analysed?

*Page 3, Line 33: the phrase allow to investigate sounds wrong to me, though Im not sure if is wrong. Better avoid anyway. Same p20/l8.

Page 12, Line 15-16: The relative difference .... Ozone sonde to MIAWARA-C...... → I think GROMOS-C is correct instead of MIAWARA-C.

Page 13, Line 14: at around 7 hPa → at around 0.07 hPa

Page 13, Line 19: averaged mean vertical wind profiles → averaged zonal mean vertical wind profiles

Page 21, Line 9: if the ozone enhancement is not linked to temperature changes, do you mean it is purely a transport effect?

**Response to comments**

1)
Page 7, Line 21: Assimilation of water in ERA5: I doubt that radiosondes and "ground stations" provide any meaningful information on stratospheric water vapour. Please comment. I noticed that later on, ERA5 is considered as a model (e.g. p11/l20, p20/l10). Would be good to find a clear perspective here.

Yes, water vapour profile measurements from radiosondes are only assimilated in the troposphere. We stated that more clearly and we consequently used the term reanalysis for ERA5 throughout

the manuscript.

For the considered time period ozone is assimilated from retrievals of the GOME-2 instruments on the METOP-A/B satellites, the SBUV-2 instruments on the NOAA satellites and of MLS and OMI on the EOS-Aura satellite up to 50 km. Water vapour is assimilated in the troposphere from humidity profiles measured with radiosondes and from ground stations which are provided by the World Meteorological Organizations Information System (WMO WIS).

2)
Page 12, Line 12: I would say the Cariolle parametrization is interactive, but certainly not a comprehensive chemistry. So the question is rather how good is the Cariolle scheme in the mesosphere?

We stated more clearly that ERA5 uses a parametrization designed for the stratosphere and added the deviation to GROMOS-C at 0.3–0.1 hPa in the mesosphere to give an estimate of how good the Cariolle scheme is in the mesosphere.

This is because ERA5 uses an ozone parametrization (Cariolle and Teyss, 2007) which is designed for the stratosphere and because there is no assimilation of ozone in the mesosphere. The difference of ERA5 relative to GROMOS-C at 0.3–0.1 hPa ranges from 100 % in summer to -50 % in winter.

3)
Page 33, the last sentence of Figure 9 caption: According to the main text in Page 13, line 35, the w_star_bar is averaged along the fit of the 5.5 ppm isopleths of SD-WACCM. So, the appropriate expression of the last sentence is The points indicate the mean of the profiles over the altitude range of isopleth of MIAWARA-C, and the stars indicate the mean along the fit of the 5.5 ppm isopleths of SD-WACCM, isnt it?

No, in Figure 9 the stars were the mean along the isopleth of MIAWARA-C. We decided to change the plot and indicate the range of the zonal mean 5.5 ppm isopleth of SD-WACCM and show the mean of $\overline{w}^*$ in this altitude range and along the isopleth. We made this change because we actually use those values for comparison with the water vapour descent rate of SD-WACCM and do not perform a direct comparison of $\overline{w}^*$ with the descent rate of MIAWARA-C.

[revised manuscript text omitted]